# Emergence of heavy tails in homogenized stochastic gradient descent

**Zhe Jiao**
School of Mathematics and Statistics
Northwestern Polytechnical University
Xi'an 710129, China
zjiao@nwpu.edu.cn

**Martin Keller-Ressel**[*]
Institute of Mathematical Stochastics
Technische Universität Dresden
01217 Dresden, Germany
martin.keller-ressel@tu-dresden.de

## Abstract

It has repeatedly been observed that loss minimization by stochastic gradient descent (SGD) leads to heavy-tailed distributions of neural network parameters. Here, we analyze a continuous diffusion approximation of SGD, called homogenized stochastic gradient descent, and show in a regularized linear regression framework that it leads to an asymptotically heavy-tailed parameter distribution, even though local gradient noise is Gaussian. We give explicit upper and lower bounds on the tail-index of the resulting parameter distribution and validate these bounds in numerical experiments. Moreover, the explicit form of these bounds enables us to quantify the interplay between optimization hyperparameters and the tail-index. Doing so, we contribute to the ongoing discussion on links between heavy tails and the generalization performance of neural networks as well as the ability of SGD to avoid suboptimal local minima.

## 1 Introduction

Stochastic gradient descent (SGD) is the cornerstone of optimization in modern deep learning (cf. [Bottou et al., 2018]). In contrast to deterministic methods, it introduces stochasticity to the optimization procedure and therefore has to be analyzed from a probabilistic viewpoint. For instance, it has been observed by Martin and Mahoney [2019], Simsekli et al. [2019], Hodgkinson and Mahoney [2021], Gurbuzbalaban et al. [2021] and others, that the distributions of neural network parameters under loss minimization by SGD are typically *heavy-tailed*. This heavy-tailed behavior has been linked to the generalization performance of neural networks: Simsekli et al. [2019] give evidence that the extreme realizations of heavy-tailed random variables allows SGD to escape local minima of the loss landscape, and Hodgkinson and Mahoney [2021] argue for a negative correlation between the parameter distributions's tail-index and the network's generalization performance.[2] For these reasons, it is important to understand the origin and effects of heavy-tailed behavior of neural network parameters in SGD. An important step in this direction has been taken in [Gurbuzbalaban et al., 2021], where the tail behavior of SGD iterates is characterized in dependence on optimization parameters, dimension and Hessian curvature at the loss minimum. One limitation of [Gurbuzbalaban et al., 2021] is that this link is described only qualitatively, but not quantitatively. Here, we provide an alternative approach through analyzing homogenized stochastic gradient descent, a diffusion approximation of SGD introduced in [Paquette et al., 2022b, Mori et al., 2022]. Leveraging Itô calculus for diffusion processes, we are able to provide more precise bounds and estimates of the tail behavior of SGD iterates, which we subsequently validate in numerical experiments.

---

[*]Center for scalable data analytics and artificial intelligence (ScaDS.ai), Leipzig/Dresden, Germany.

[2]The tail-index is a quantitative measure of heavy-tailedness, with a smaller tail index indicating increased heaviness of tails; see Section 2.4. See also [Raj et al., 2023, Benjamin and Simsekli, 2024] for further results on the connection between generalization and heavy tails.

38th Conference on Neural Information Processing Systems (NeurIPS 2024).

## 1.1 Our contribution

Our contribution to the analysis of heavy-tailed phenomena in SGD can be summarized as follows:

- We introduce a new method, namely comparison results in *convex stochastic order* for homogenized stochastic gradient descent. These comparison results, given in Section 3 allow us to link SGD to the well-studied class of *Pearson Diffusions* (cf. [Forman and Sørensen, 2008]) and then to obtain bounds for their tail-index.

- Contrary to [Gurbuzbalaban et al., 2021], who describe the tail-index only implicitly (observing phase-transitions between different regimes) our tail-index bounds are fully explicit. Moreover, their explicit form is validated in numerical experiments in Section 4.

- Our results suggest (skew) Student-$t$-distributions as surrogate for parameter distributions in neural networks under SGD, in contrast to the earlier work of [Gurbuzbalaban et al., 2021] where $\alpha$-stable distributions have been suggested. This proposal is validated by numerical experiments and statistical test in Section 4.

- Finally, our results challenge the claim that the *'observed heavy-tailed behavior of SGD in practice cannot be accurately represented by an SDE driven by a Brownian motion'* put forward in [Simsekli et al., 2020]. Our modeling approach is based on hSGD – an SDE driven by Brownian motion – which asymptotically exhibits heavy-tailed behavior with a tail-index that, in experiments, closely matches the empirical tail index of SGD iterates on real data.

## 2 Background

### 2.1 Empirical risk minimization

The general framework for training deep neural networks is to solve the problem of empirical risk minimization

$$\min_{x\in\mathbb{R}^d}\left\{L(x):=\frac{1}{n}\sum_{i=1}^{n}L_i(x)\right\},\tag{ERM}$$

where $L_i$ denotes the loss induced by the data point $a_i \in \mathbb{R}^d$ with label/response $b_i \in \mathbb{R}$, given the model's parameter vector $x \in \mathbb{R}^d$. For our theoretical and numerical analysis of heavy-tailed phenomena we focus on the specific case of regularized linear regression. Hence, as in [Gurbuzbalaban et al., 2021], we assume a quadratic structure of $L_i(x)$, setting

$$L_i(x) = \frac{1}{2}(a_i \cdot x - b_i)^2.$$

Including a regularization term weighted by $\delta \geq 0$, we arrive at the objective function

$$L^{\text{reg}}(x) = L(x) + \frac{\delta}{2n}|x|^2 = \frac{1}{n}\left(\sum_{i=1}^{n}L_i(x) + \frac{\delta}{2}|x|^2\right),\tag{$\delta$-ERM}$$

which is the loss function of *ridge regression* (cf. [Hastie et al., 2009]). We arrange the training data into a design matrix $A \in \mathbb{R}^{n \times d}$ and label vector $b \in \mathbb{R}^n$, whose $i$-th row are given by $a_i$ and $b_i$ respectively, allowing the write ($\delta$-ERM) as s

$$L^{\text{reg}}(x) = \frac{1}{2n}|Ax - b|^2 + \frac{\delta}{2n}|x|^2$$

with gradient given by $\nabla L^{\text{reg}}(x) = \frac{1}{n}\left(A^\top(Ax - b) + \delta x\right)$.

### 2.2 Stochastic gradient descent

The standard approach to solve the problem of empirical risk minimization in deep learning is to use stochastic gradient descent (SGD) or any of its generalizations involving momentum, adaptive learning rates, gradient rescaling, etc. (cf. [Goodfellow et al., 2016, Bottou et al., 2018]). As a first step, we consider plain SGD with constant learning rate $\gamma$, which can be written in recursive form as

$$x_{k+1} = x_k - \gamma\nabla L^{\text{reg}}_{\Omega_k}(x_k),\tag{SGD}$$

where $\nabla L_{\Omega_k}^{\mathrm{reg}}(x_k) = \frac{1}{B}\sum_{i\in\Omega_k} L_i^{\mathrm{reg}}(x)$ and $\Omega_k$ is a batch of size $B \geqslant 1$ sampled uniformly and independently from $\{1,\cdots,n\}$. It will be convenient to rewrite (SGD) as

$$x_{k+1} = x_k - \gamma\nabla L^{\mathrm{reg}}(x_k) + \gamma\varepsilon(x_k), \tag{1}$$

where the gradient noise is given by

$$\varepsilon(x_k) = -[\nabla L_{\Omega_k}(x_k) - \nabla L(x_k)]. \tag{2}$$

Note that the gradient noise is unbiased (i.e. $\mathbb{E}\varepsilon(x) = 0$) with covariance matrix given by[3]

$$C(x) := \mathbb{E}\left[\varepsilon(x)\varepsilon(x)^\top\right] = \frac{1}{B}\left(\frac{1}{n}\sum_{i=1}^n \nabla L_i(x)\nabla L_i(x)^\top - \frac{1}{n^2}\nabla L(x)\nabla L(x)^\top\right).$$

The theoretical properties of SGD can now be either analysed directly through the stochastic recurrence (1) (cf. [Bottou et al., 2018]) or through a continuous diffusion approximation, known in the general case as *stochastic modified equation*, cf. [Mandt et al., 2016, Li et al., 2017]. This approximation is obtained by recognizing (1) as the Euler-Maruyama approximation (in the small learning-rate regime) of the stochastic differential equation (SDE)

$$dX_t = -\gamma\nabla L^{\mathrm{reg}}(X_t)dt + \gamma\sqrt{C(X_t)}dW_t \tag{SME}$$

driven by a $d$-dimensional Brownian motion $(W_t)_{t\geq 0}$; cf. Thm. 1 in [Li et al., 2017]. A common further simplification is to assume that the covariance matrix $C(x)$ is constant, yielding the Ornstein-Uhlenbeck-approximation (also known as Langevin equation) of SGD, cf. [Mandt et al., 2016, Li et al., 2017].

### 2.3 Homogenized Stochastic Gradient Descent

Our analysis of SGD is based on *homogenized stochastic gradient descent* (hSGD), introduced concurrently in [Paquette et al., 2022a] and [Mori et al., 2022], which is another approximation of (SME). In contrast to the Ornstein-Uhlenbeck-approximation where the covariance matrix of gradient noise is assumed constant, hSGD uses the more elaborate 'decoupling approximation'

$$C(x) \approx \frac{2}{B}L(x)\nabla^2 L(x),$$

see [Paquette et al., 2022a] and [Mori et al., 2022] for a derivation. Hence, in our notation, hSGD for penalized empirical risk minimization is given by[4]

$$dX_t = -\gamma\nabla L^{\mathrm{reg}}(X_t)dt + \gamma\sqrt{\frac{2}{B}L(X_t)\nabla^2 L(X_t)}dW_t. \tag{hSGD}$$

In the regime where $n$ and $d$ are simultaneously large, and under certain assumptions on the distribution of the data $A$ and $b$, [Paquette et al., 2022a] provide approximation guarantees of the following form: For any given $T > 0$ and $D > 0$, there is a $C > 0$, such that

$$\mathbb{P}\left(\sup_{0\leq t\leq T}\left|\mathcal{R}(x_{\lfloor tn\rfloor}) - \mathcal{R}(X_t)\right| > d^{-\epsilon/2}\right) \leq Cd^{-D} \tag{3}$$

for quadratic statistics $\mathcal{R} : \mathbb{R}^d \to \mathbb{R}$ and when $n \geq d^\epsilon$ for some $\epsilon > 0$; cf. Thm. 1.3 in [Paquette et al., 2022a] for details. Further empirical evidence for the approximation quality of hSGD with respect to SGD can is also given in [Paquette et al., 2022a, Mori et al., 2022], altogether providing a sufficient basis for analyzing the properties of SGD through hSGD.

Furthermore, the stochastic differential equation (hSGD) can be simplified by using the reduced singular value decomposition (SVD) of the design matrix $A$. In detail, let $r = \mathrm{rank}(A) \leqslant d$, and let $A = P\Sigma Q^\top$ be the reduced SVD of $A$, where $Q$ is $d$-by-$r$ and satisfies $Q^\top Q = I_r$, $P$ is $n$-by-$r$ and satisfies $P^\top P = I_r$, and

$$\Sigma = \mathrm{diag}\{\lambda_j\}, \quad \lambda_1 \geqslant \lambda_2 \geqslant \cdots \geqslant \lambda_r > 0$$

is the diagonal matrix of non-zero singular values of $A$. We distinguish the following two cases of hSGD:

---

[3]Full derivation given in Supplement A.1.

[4]We remark that Paquette et al. [2022a] assume a batch size of $B = 1$; the derivation of [Mori et al., 2022], however, does not restrict $B$.

- *Underparametrized hSGD*: $Ax = b$ has no exact solution,
- *Overparametrized hSGD*: $Ax = b$ has an exact solution,

and impose the following assumption:

**Assumption 2.1.** In the overparametrized case, we require $\delta > 0$, i.e. the loss function must be regularized.

It is easily verified that $x_* = Q\Sigma^{-1}P^\top b$ is the unique global minimum of the unregularized loss in the underparametrized case and the global minimum of smallest norm in the overparametrized case. We set $Y_t = (Y_t^i)_{i=1}^r = Q^\top X_t - Q^\top x_*$ and obtain the following system of SDEs[5] for the 'centered principal components' $(Y_t^1, \ldots, Y_t^r)$ of (hSGD)

$$dY_t^i = -\frac{\gamma}{n}\left[\left(\lambda_i^2 + \delta\right)Y_t^i - \delta\alpha_i\right]dt + \frac{\lambda_i\gamma}{n}\sqrt{\frac{1}{B}\left[\sum_{j=1}^r(\lambda_j Y_t^j)^2 + \beta\right]}\,dB_t^i \tag{4}$$

with

$$\alpha = (\alpha_i)_{i=1}^r = -\Sigma^{-1}P^\top b, \qquad \beta = b^\top(I_n - PP^\top)b \geq 0$$

and $(B_t)_{t\geq 0}$ a $r$-dimensional Brownian motion, obtained as an orthogonal transformation $B_t = Q^\top W_t$ of the $d$-dimensional Brownian motion $(W_t)_{t\geq 0}$. Note that $PP^\top b$ is the projection of $b$ onto the column space of $A$. Thus, in the overparametrized case, $PP^\top b = b$ and hence $\beta = 0$, whereas in the underparametrized case $PP^\top b \neq b$ and hence $\beta > 0$. Here, our main objective is to use (hSGD) to study the distributional properties, in particular the tail behavior, of SGD iterates.

## 2.4 Heavy-Tailed Distributions

We collect some relevant definitions related to heavy-tailed distributions and their tail index (cf. [Resnick, 2007, Foss et al., 2011]).

**Definition 2.2** (See Def. 1.1 in Foss et al. [2011])**.** A distribution function $F(z)$ is said to be *heavy-tailed* (at the right end) if and only if

$$\limsup_{z \to \infty} \frac{1 - F(z)}{e^{-sz}} = \infty, \quad \text{for all } s > 0.$$

A real-valued random variable is said to be heavy-tailed if its distribution function is heavy-tailed.

**Definition 2.3.** An $\mathbb{R}^d-$valued random vector $X$ is heavy-tailed if $u^{\mathrm{T}}X$ is heavy-tailed for some vector $u \in \mathbb{S}^{d-1} := \{u \in \mathbb{R}^d : |u| = 1\}$.

**Definition 2.4.** The *tail-index* of an $\mathbb{R}^d-$valued random vector $X$ is defined as

$$\eta := \sup\{p \geq 0 : \mathbb{E}[|X|^p] < \infty\} \in [0, \infty].$$

In particular, a finite tail-index $\eta < \infty$ implies heavy-tailedness of $X$, and lower values of $\eta$ signify increased heaviness of tails and more extremal behavior. A tail index of $\eta < 2$, for example, implies infinite variance and $\eta < 1$ implies non-existence of even the mean of $X$. Examples of heavy-tailed distributions are the lognormal distribution, the Student-$t$-distribution, the Pareto (power-law) distribution, and $\alpha$-stable distributions.

Finally, we introduce a definition related to the asymptotic behavior of stochastic processes.

**Definition 2.5.** Let $X = (X_t)_{t\geq 0}$ be a stochastic process. The *asymptotic tail-index* of $X$ is defined as

$$\eta := \sup\{p \geq 0 : \limsup_{t\to\infty}\mathbb{E}[|X_t|^p] < \infty\}. \tag{5}$$

---

[5]Full derivation given in Supplement A.2.

## 2.5 Pearson Diffusions

To analyze its tail behavior, we perform a further rescaling of (4) by setting, for $i \in \{1, \ldots, r\}$,

$$Z_t^i = \begin{cases} \lambda_i \mathrm{sign}(\alpha_i) Y_t^i, & \beta = 0, \\ \frac{\lambda_i}{\sqrt{\beta}} Y_t^i, & \beta > 0, \end{cases} \quad \mu_i = \begin{cases} \frac{n\lambda_i |\alpha_i|}{\lambda_i^2 + \delta}, & \beta = 0, \\ \frac{n\lambda_i \alpha_i}{\sqrt{\beta}(\lambda_i^2 + \delta)}, & \beta > 0, \end{cases} \quad \chi = \begin{cases} 0, & \beta = 0, \\ 1, & \beta > 0, \end{cases} \tag{6}$$

$$\theta_i = \frac{\gamma}{n}\left(\lambda_i^2 + \delta\right) > 0 \quad \text{and} \quad \phi_i = \frac{\gamma \lambda_i^4}{2nB(\lambda_i^2 + \delta)} > 0.$$

This recasts the system (4) to

$$dZ_t^i = -\theta_i(Z_t^i - \mu_i)dt + \sqrt{2\theta_i \phi_i(|Z_t|^2 + \chi)}dB_t^i \tag{7}$$

with $|Z_t|^2 = \sum_{i=1}^r (Z_t^i)^2$. These SDEs now have a clear structural resemblance to the system of independent one-dimensional SDEs

$$d\hat{Z}_t^i = -\theta(\hat{Z}_t^i - \mu_i)dt + \sqrt{2\theta_i \phi_i((\hat{Z}_t^i)^2 + \chi)}dB_t^i \tag{8}$$

with the only difference given by the coupling of (7) through the $|Z_t|^2$-term in the diffusion coefficient.[6] The components of (8) are independent *Pearson diffusions*. Pearson diffusions are a flexible class of SDEs with a unified theory for statistical inference and with stationary distributions known as Pearson distributions (cf. [Forman and Sørensen, 2008]). In more detail, we obtain from [Forman and Sørensen, 2008] the following properties:

*Underparametrized hSGD ($\beta > 0$):* $\hat{Z}_t^i$ is $\mathbb{R}$-valued and the stationary distribution of $\hat{Z}_t^i$ is called Pearson's type IV distribution (or skew Student $t$-distribution) and has the unnormalized density

$$p_i(u) \propto \left[1 + \left(\frac{u}{\sqrt{\nu_i}} + \mu_i\right)^2\right]^{-\frac{\nu_i + 1}{2}} \exp\left\{\mu_i(\nu_i - 1)\arctan\left(\frac{u}{\sqrt{\nu_i}} + \mu_i\right)\right\} \tag{9}$$

with $\nu_i = \phi_i^{-1} + 1$.

*Overparametrized hSGD ($\beta = 0$):* $\hat{Z}_t^i$ is $(0, \infty)$-valued and the stationary distribution of $\hat{Z}_t^i$ is called Pearson's type V distribution (or inverse Gamma distribution) and has the unnormalized density

$$p_i(u) \propto u^{-\nu_i - 1} \exp\left(-\frac{\mu_i(\nu_i - 1)}{u}\right) \tag{10}$$

with $\nu_i = \phi_i^{-1} + 1$.

In both cases, the stationary distribution is heavy-tailed with tail-index given by $\nu_i$, thus providing a first connection between the SDE-approach and the emergence of heavy-tails. This connection will be quantified and made rigorous in Section 3.

## 2.6 Comparison to existing literature

We compare our approach to studying the distributional properties of SGD through (hSGD) with other continuous-time approximations: The Ornstein-Uhlenbeck-approximation uses (SME) under the additional assumption that the covariance matrix $C(x)$ is constant. Thus, gradient noise is approximated by Gaussian noise and the Gaussian noise enters (SME) *additively*. The $\alpha$-stable Ornstein-Uhlenbeck-approximation of [Gurbuzbalaban et al., 2021] instead presumes (based on a generalized central limit theorem) that gradient noise is non-Gaussian and follows an $\alpha$-stable law. Moreover, the noise is assumed state-independent, and therefore also enters additively. In (hSGD), gradient noise is locally (i.e., conditionally on the state $X_t$) Gaussian, but *state-dependent*. The diffusion term in (7) reveals that the noise enters the SDE both multiplicatively (through the $|Z_t|^2$-term) and additively (through the constant $\chi$). Moreover, $\chi = 0$ in the overparametrized case, such that we observe a phase transition from a mix of additive and multiplicative noise in the underparametrized case, to purely mulitiplicative noise in the overparametrized case. We note that the importance of multiplicative noise in models of SGD dynamics is discussed in great detail in [Hodgkinson and Mahoney, 2021]. We provide a summary of the comparison of these approaches in Table 1

---

[6]Existence and uniqueness of the solutions to these SDEs follows from standard results, cf. [Karatzas and Shreve, 2014, Ch. 5, Thm. 2.5] or Oksendal [2013].

Table 1: Comparison of continuous-time models of SGD

| Model | local gradient noise | global parameter distribution | tail-index |
|---|---|---|---|
| Gaussian OU | Gaussian additive | Gaussian | $+\infty$ |
| $\alpha$-stable OU | Non-Gaussian additive | Non-Gaussian ($\alpha$-stable) | $(0, 2)$ |
| homogenized SGD | Gaussian additive/multiplicative | Non-Gaussian (with Student-$t$ as proxy) | $(1, \infty)$ |

## 3 Theoretical results

### 3.1 Moment comparison

Our first result shows that the decoupled Pearson diffusions (8) are lower bounds, in *convex stochastic order*[7], to the coupled hSGD process (7). In particular, a comparison result for moments holds.

**Theorem 3.1.** *For $i = 1, \cdots, d$, let $(Z_t^i)_{t \geqslant 0}$ be the components of the rescaled* (hSGD) *from* (7) *and $(\hat{Z}_t^i)_{t \geqslant 0}$ be the independent Pearson diffusion from* (8). *Then for any $t \geqslant 0$ and convex function $g : \mathbb{R} \to \mathbb{R}$ it holds that*

$$\mathbb{E}[g(Z_t^i)] \geq \mathbb{E}[g(\hat{Z}_t^i)]. \tag{11}$$

*In particular this implies the ordering of $p$-moments*

$$\mathbb{E}[|Z_t^i|^p] \geq \mathbb{E}[|\hat{Z}_t^i|^p] \tag{12}$$

*for all $p \geq 1$.*

Note that finiteness of the expectations does not need to be assumed, i.e., the inequalities also hold if one of the expectations takes the value $+\infty$. Comparison results for SDEs generally require two conditions (cf. [Bergenthum and Rüschendorf, 2007]): An ordering between the drift- and diffusion-coefficients of the two SDEs, and the 'propagation-of-order'-property for one of the processes. Comparing (7) and (8), we see that the drift coefficients are identical, while the diffusion coefficients satisfy the required ordering condition $2\theta_i \phi_i(|z|^2 + \chi) \geq 2\theta_i \phi_i(z_i^2 + \chi)$ for any $z \in \mathbb{R}^r$. The propagation-of-order property of $\hat{Z}$ and the full proof of Theorem 3.1 are provided in Supplement A.3.

### 3.2 Upper and lower bounds for the asymptotic tail index

Since the process $(Z_t)_{t \geq 0}$ is a linear transformation of the hSGD process $(X_t)_{t \geq 0}$, it is clear that the tail behaviour of their marginal distributions – in particular the finiteness of $p$-moments – is identical. Hence, an application of Thm. 3.1 provides an upper bound on the asymptotic tail index of (hSGD):

**Theorem 3.2.** *The asymptotic tail index $\eta$ of* (hSGD) *has the upper bound*

$$\eta \leq \eta^* := 1 + \frac{2nB(\lambda_1^2 + \delta)}{\gamma \lambda_1^4}. \tag{13}$$

Under conditions on the learning rate $\gamma$, a complementary lower bound can be derived from existing results on moment stability of SDEs, see Thm. 5.2 in [Li et al., 2019] and Supplement A.5 for details:

**Theorem 3.3.** *Suppose that the learning rate $\gamma$ satisfies*

$$\gamma < \overline{\gamma} =: \frac{2nB(\lambda_1^2 + \delta)}{\lambda_1^2 \sum_{i=1}^r \lambda_i^2},$$

*then the asymptotic tail index $\eta$ of* (hSGD) *has the lower bound*

$$\eta \geq \eta_* := 1 + \frac{2nB(\lambda_1^2 + \delta)}{\gamma \lambda_1^4} - \frac{\sum_{i=2}^r \lambda_i^2}{\lambda_1^2}. \tag{14}$$

### 3.3 Wasserstein convergence

Theorems 3.2 and 3.3 are results on the *asymptotic* tail index, raising the question how fast convergence to the stationary distribution takes place. The next result shows that, under a suitable assumption on the learning rate, convergence takes place exponentially fast in 2-Wasserstein distance:

---

[7]See e.g. [Shaked and Shanthikumar, 2007]

**Theorem 3.4.** *Suppose that the learning rate $\gamma$ satisfies*

$$\gamma < \gamma' =: \frac{nB}{2} \left\{ \sum_{i=1}^{r} \frac{\lambda_i^4}{\lambda_i^2 + \delta} \right\}^{-1}.$$

*Then the equation*

$$\sum_{i=1}^{r} \frac{\lambda_i^4}{\lambda_i^2 + \delta - n\rho/\gamma} = \frac{nB}{2\gamma}$$

*has a unique positive solution $\rho_* > 0$, and the marginal distribution $\pi_t$ of the hSGD process $X_t$ converges in 2-Wasserstein distance $\mathcal{W}_2$ to its unique invariant distribution $\pi$. Moreover, there exists $C > 0$, such that*

$$\mathcal{W}_2(\pi_t, \pi) \le Ce^{-t\rho_*}.$$

We remark that if the conditions of Theorem 3.4 are satisfied, then the asymptotic tail-index $\eta$ is necessarily greater than two, such that second moments and in particular the 2-Wasserstein distance are well-defined and finite.

### 3.4 Discussion of theoretical results

We compare our results to Gurbuzbalaban et al. [2021], who analyse the distributional properties of SGD directly through the stochastic recurrence (1) under the assumption of an isotropic Gaussian data distribution. In our setting, the data distribution is arbitrary, since all results are given conditional on the data matrix $A$. On the other hand, we analyse SGD only through its diffusion approximation (hSGD) rather than directly. However, in contrast to [Gurbuzbalaban et al., 2021], we obtain the *quantitative* and *explicit* tail-index bounds (13) and (14), whereas Gurbuzbalaban et al. [2021] only describe the tail index through an *implicit equation* and derive *qualitative results* on its behaviour.

*Parameter Dependency.* Some interesting observations can be made when we consider the dependency of $\eta$ on several meta-parameters of the stochastic gradient descent procedure:

**Corollary 3.5.** *The upper and lower bounds of the tail-index are increasing in the regularization parameter $\delta$ and batch size $B$, and are decreasing in the learning rate $\gamma$ and the first singular value $\lambda_1$ of the data matrix $A$.*

This result agrees with Theorem 4 in [Gurbuzbalaban et al., 2021], obtained under the assumption of an isotropic data distribution $a_i \sim N(0, \sigma^2 I_d)$, in all aspects, except the dependency on dimension $d$. While Gurbuzbalaban et al. [2021] report decreasing dependency on $d$, our tail-index bounds do not explicitly depend on dimension $d$. Nevertheless, the two results can be reconciled as follows: Under the assumptions in [Gurbuzbalaban et al., 2021], the data matrix $A = (a_i)$ is random with $\mathbb{E}(A^{\mathrm{T}}A) = \sigma^2 I_d$, and the product matrix $W := A^{\mathrm{T}}A$ follows the so-called Wishart ensemble (cf. [Wishart, 1928]). Moreover, from Theorem 1.1 in [Johnstone, 2001] it follows that for large $d$ the maximum eigenvalue of $W$ is

$$\lambda_1^2 = \sigma^2 \left[ \left( \frac{1}{\sqrt{r}} + 1 \right)^2 d + r^{\frac{1}{6}} \left( \frac{1}{\sqrt{r}} + 1 \right)^{\frac{4}{3}} d^{\frac{1}{3}} \Psi \right], \tag{15}$$

where the ratio $r = \frac{d}{n-1} < 1$ and the distribution function of the random variable $\Psi$ is the well-known Tracy-Widom distribution of order 1 (cf. [Craig and Harold, 1996]). From (15), we can calculate the average of $\lambda_1^2$ as

$$\mathbb{E}\left[\lambda_1^2\right] = \sigma^2 (\frac{1}{\sqrt{r}} + 1)^2 d = \sigma^2 (\sqrt{n-1} + \sqrt{d})^2$$

and $\lambda_1^2$ fluctuates around this expectation over a narrow region of width $O(d^{\frac{1}{3}})$. Substituting $\lambda_1^2$ by its expectation in (13) and (14) we can now see that $\eta_*$ and $\eta^*$ increase in both variance $\sigma^2$ and $d$, consistent with [Gurbuzbalaban et al., 2021].

*Distributional properties.* From Theorem 3.1 we see that the skew Student-$t$ distribution provides an asymptotic lower bound in convex order for the marginal distribution of hSGD. Empirically (see Section 4) we see that skewness is negligible and furthermore, that the Student-$t$-distribution not only provides a lower bound, but in fact a very good fit to the parameter distribution of SGD in general, surpassing the fit of the $\alpha$-stable distribution proposed in [Gurbuzbalaban et al., 2021]. For this reason, we propose to use the Student-$t$-distribution, rather than $\alpha$-stable distribution, as a proxy for the parameter distribution in SGD.

# 4 Experiments

Based on the upper and lower bounds in Theorems 3.2 and 3.3, we present some experiments to illustrate the tail behavior of SGD and the factors influencing the tail index. The procedure of our experiments contains the following steps.

1. Given $[\text{data}|b]$, we transform the data to be on a similar scale by the linear scaling
$$A = \frac{\text{data} - \min\{\text{data}\}}{\max\{\text{data}\} - \min\{\text{data}\}}.$$

2. Let $K$ be the iteration number of SGD. We apply (SGD) to solve (ERM). The final state $x_K \in \mathbb{R}^d$ is a random vector.

3. Repeat the second step 1000 times for different initial points and obtain 1000 different samples of $x_K$.

4. For further distributional analysis we project $x_K$ via $y = q_1^\top x_K$ on the dominant direction, given by the first right singular vector $q_1$ of $A$. Then we utilize the 1000 samples to obtain the empirical complementary cumulative distribution function (ccdf) of y.

## 4.1 Datasets

*Synthetic data.* We first validate our results in the same synthetic setup used in [Gurbuzbalaban et al., 2021]. All data points are drawn from isotropic Gaussian distributions, precisely, the $i$-th row of $\mathcal{X} \in \mathbb{R}^{n \times d}$ contains $\chi_i \in \mathbb{R}^d \sim \mathcal{N}(0, I_d)$. Then given $x \in \mathbb{R}^d \sim \mathcal{N}(0, 3I_d)$ we draw the response vextor $b \in \mathbb{R}^n$ with components $b_i \sim \mathcal{N}(\chi_i x, 3)$. We set the number $n$ of the synthetic data to be 2000 through our experiments.

*Real data.* In our second setup we conduct our experiments on the handwritten digits dataset from the Scikit-learn python package (cf. [Pedregosa et al., 2011]) using a random feature model proposed in [Rahimi and Recht, 2007] and a three-layer neural network. The digits dataset contains $n = 1797$ images of handwritten digits in a $8 \times 8$ pixel format. The pixels are stacked into vectors of length $n_0 = 8^2 = 64$ resulting in a raw data matrix $\mathcal{Y} \in \mathbb{R}^{n \times n_0}$ and the class label $b_i = \{0, 1, \cdots, 9\}$ is used as response vector. For the random feature model, we choose a dimension $d$ and draw a random weight matrix $W \in \mathbb{R}^{n_0 \times d}$ having standard Gaussian entries. The feature matrix $W \in \mathbb{R}^{n \times d}$ is given by
$$\mathcal{Z} = \sigma\left(\frac{\mathcal{Y}W}{\sqrt{n_0}}\right) \in \mathbb{R}^{n \times d},$$
where $\sigma(\cdot)$ is a rescaled ReLu activation function. The neural-network model uses 64 neurons in each hidden layer and sigmoid activation functions. The precise parameter values used for the figures are reported in Tables 4 and 5 in the supplement.

## 4.2 Empirical results

To verify the heavy-tailed behavior of y as well as our tail-index bounds from Theorems 3.2 and 3.3 and the distributional approximation suggested by (9), we use MLE-estimation to fit our centered data as
$$z := y - \text{mean}\{y\} \sim \kappa t(\nu),$$
where $t(\nu)$ denotes a Student-$t$-distribution with parameter $\nu$ and $\kappa$ is a fitted scaling factor.[8] The QQ-plots in Figures 1, 2 (a)-(c) show that the Student-$t$-distribution provides a very good fit to the empirical data, validating our use of Pearson diffusions to approximate SGD. In comparison, it can be seen in Figure 1, 2 (d)-(f) that the fitted $\alpha$-stable distribution overestimates the heaviness of tails, in particular for the random feature model on real data. We complement these figures by Kolmogorov-Smirnov tests (cf. Chapter 4.4 in [Corder and Foreman, 2014]) testing for the goodness-of-fit of the Student-$t$-distribution and the $\alpha$-stable distribution respectively; see Tables 2, 3 for detailed results. In all three settings, the hypothesis of a Student-$t$-distribution is accepted, while the $\alpha$-stable distribution is rejected.

---

[8]Eq. (9) actually implies a skew Student-$t$-distribution, but we use a symmetric one to avoid the estimation of an additional parameter $\mu$.

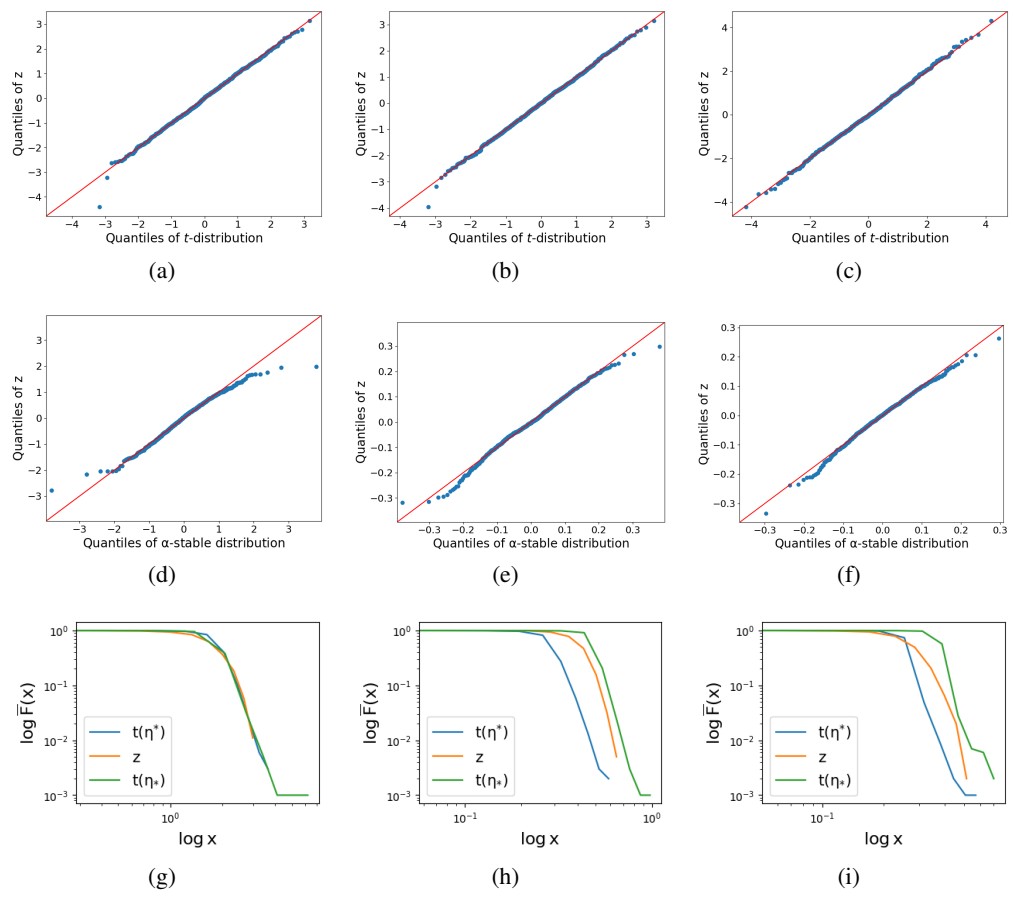

Figure 1: Results for linear regression/random feature model trained on datasets $\mathcal{X}$, $\mathcal{Y}$, and $\mathcal{Z}$. (a)-(c) Quantile-Quantile plots of fitted Student-$t$-distribution against empirical SGD iterates; (d)-(f) Quantile-Quantile plots of fitted $\alpha$-stable distribution against empirical SGD iterates; (g)-(i) Comparison between ccdf of empirical data and Student-$t$-distribution parameterized by upper tail-index bound $\eta^*$ and lower bound $\eta_*$.

Moreover, in Figure 1 (g)-(i) we plot (in doubly logarithmic coordinates) the empirical ccdf of the SGD iterates z, together with the ccdf of the Student-$t$-distribution parametrized by our lower and upper bound $\eta_*$ and $\eta^*$. It can be seen that the empirical ccfd, including its tail, is nicely sandwiched between upper and lower bound, validating Theorems 3.2 and 3.3. Additionally, we once more confirm the heavy-tailed behavior of SGD iterates as already observed in [Simsekli et al., 2019, Hodgkinson and Mahoney, 2021, Gurbuzbalaban et al., 2021].

Table 2: Kolmogorov-Smirnov test of theoretical distributions against observed SGD iterates of the linear regression/random feature model. The null hypothesis $H_0$ is that two distributions are identical, the alternative $H_1$ is that they are not identical.

| Distribution | Dataset | In Fig. 1 | K-S statistic | $p$-value | decision |
|---|---|---|---|---|---|
| Student-$t$ | $\mathcal{X}$ | (a) | 0.029 | $0.795 > 0.05$ | accept $H_0$ |
| Student-$t$ | $\mathcal{Y}$ | (b) | 0.039 | $0.433 > 0.05$ | accept $H_0$ |
| Student-$t$ | $\mathcal{Z}$ | (c) | 0.030 | $0.759 > 0.05$ | accept $H_0$ |
| $\alpha$-stable | $\mathcal{X}$ | (d) | 0.084 | $0.002 < 0.05$ | reject $H_0$ |
| $\alpha$-stable | $\mathcal{Y}$ | (e) | 0.067 | $0.022 < 0.05$ | reject $H_0$ |
| $\alpha$-stable | $\mathcal{Z}$ | (f) | 0.070 | $0.015 < 0.05$ | reject $H_0$ |

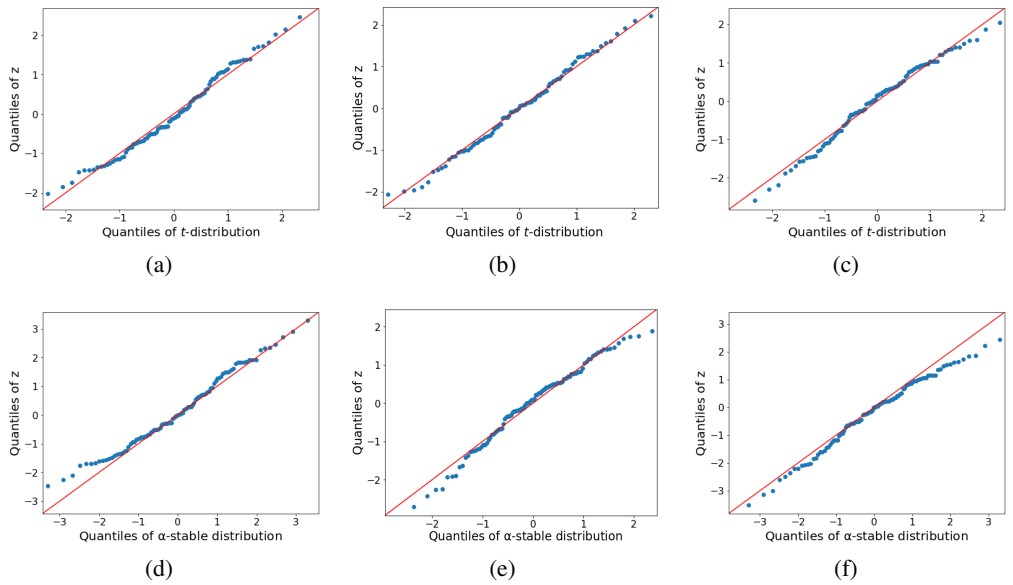

Figure 2: Results for three-layer neural network model trained on datasets $\mathcal{X}$, $\mathcal{Y}$, and $\mathcal{Z}$. (a)-(c) Quantile-Quantile plots of fitted Student-$t$-distribution against empirical SGD iterates of second layer; (d)-(f) Quantile-Quantile plots of fitted $\alpha$-stable distribution against empirical SGD iterates of second layer.

Table 3: Kolmogorov-Smirnov test of theoretical distributions against observed SGD iterates from the second layer of the three-layer neural network model. The null hypothesis $H_0$ is that two distributions are identical, the alternative $H_1$ is that they are not identical.

| Distribution | Dataset | In Fig. 2 | K-S statistic | $p$-value | decision |
|---|---|---|---|---|---|
| Student-$t$ | $\mathcal{X}$ | (a) | 0.011 | $0.208 > 0.05$ | accept $H_0$ |
| Student-$t$ | $\mathcal{Y}$ | (b) | 0.061 | $0.871 > 0.05$ | accept $H_0$ |
| Student-$t$ | $\mathcal{Z}$ | (c) | 0.060 | $0.883 > 0.05$ | accept $H_0$ |
| $\alpha$-stable | $\mathcal{X}$ | (d) | 0.078 | $0.001 < 0.05$ | reject $H_0$ |
| $\alpha$-stable | $\mathcal{Y}$ | (e) | 0.084 | $0.035 < 0.05$ | reject $H_0$ |
| $\alpha$-stable | $\mathcal{Z}$ | (f) | 0.070 | $0.015 < 0.05$ | reject $H_0$ |

## 5   Conclusion and Limitations

This study delves into the phenomenon of heavy tails emerging in the parameters of homogenized stochastic gradient descent applied to regularized linear regression. By establishing a connection between hSGD and Pearson diffusions, we have been able to derive both explicit upper and lower bounds for the tail index of the parameter distribution. Our results reveal that heavy tails can emerge even in the presence of locally Gaussian gradient noise and provide insights into the influence of optimization hyperparameters on the tail index. However, it is essential to recognize that our analysis relies on the approximation of SGD by hSGD and is limited to the setting of linear regression with quadratic loss. Another limitation (see (14)) is that the tail-index of hSGD is lower-bounded by one, and thus hGSD can not be used to analyse 'ultra-heavy tails' with tail-index $\eta \leq 1$. Future work will be devoted to extending our results to non-linear models and to providing a tighter connection between the behaviour of hSGD and the discrete-time SGD algorithm.

## Acknowledgments and Disclosure of Funding

Zhe Jiao's research is supported by the National Natural Science Foundation of China (12272297). Martin Keller-Ressel acknowledges support from the Center for Scalable Data Analytics and Artificial Intelligence (ScaDS.AI) in Lepizig/Dresden, Germany.

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

# A Supplementary material

## A.1 Derivation of covariance matrix

Consider the minibatch stochastic gradient

$$\nabla L_k(x) = \frac{1}{B} \sum_{i \in \Omega_k} L_i(x) = \frac{1}{B} \sum_{i \in \Omega_k} \nabla L_i(x),$$

where $B$ is the batchsize and the random set $\Omega_k = \{i_1, \cdots, i_B\}$ consists of $B$ independently identically distributed random integers sampled uniformly from $\{1, 2, \cdots, n\}$.

Let $\nabla \tilde{L}_k(x) = \frac{1}{B} \sum_{i \in \Omega_k} \nabla L_i(x)$. It can be rewritten as

$$\nabla \tilde{L}_k(x) = \frac{1}{B} \sum_{i=1}^n \nabla L_i(x) s_i,$$

where the random variable $s_i = l$ if $l$-multiple $i$'s are sampled in $\Omega_k$, with $0 \leqslant l \leqslant B$. The probability of $s_i = l$ is given by the multinomial distribution $\mathbb{P}(s_i = l) = C_B^l (\frac{1}{n})^l (1 - \frac{1}{n})^{B-l}$. Moreover, we have

$$\mathbb{E}[s_i] = \frac{B}{n}, \quad \mathbb{E}[s_i s_j] = \frac{B(B-1)}{n^2}, \quad \mathbb{E}[s_i s_i] = \frac{Bn + B(B-1)}{n^2}.$$

We can also compute

$$\mathbb{E}[\nabla \tilde{L}_k(x)] = \frac{1}{B} \sum_{i=1}^n \nabla L_i(x) \mathbb{E}[s_i] = \frac{1}{n} \nabla L(x) \tag{16}$$

and

$$\mathbb{E}[\nabla \tilde{L}_k(x) \nabla \tilde{L}_k(x)^\top]$$

$$= \frac{1}{B^2} \mathbb{E}\left[\sum_{i=1}^n \sum_{j=1}^n \nabla L_i(x) \nabla L_j(x)^\top s_i s_j\right] = \frac{1}{B^2} \sum_{i=1}^n \sum_{j=1}^n \left[\nabla L_i(x) \nabla L_j(x)^\top \mathbb{E}(s_i s_j)\right]$$

$$= \frac{1}{B^2} \sum_{i,j=1}^n \nabla L_i(x) \nabla L_j(x)^\top \frac{B(B-1)}{n^2} \tag{17}$$

$$+ \frac{1}{B^2} \sum_{i=1}^n \nabla L_i(x) \nabla L_i(x)^\top \left[\frac{Bn + B(B-1)}{n^2} - \frac{B(B-1)}{n^2}\right]$$

$$= \frac{B-1}{B} \frac{1}{n^2} \nabla L(x) \nabla L(x)^\top + \frac{1}{nB} \sum_{i=1}^n \nabla L_i(x) \nabla L_i(x)^\top.$$

Combining (16) with (17) gives

$$C(x) = \mathbb{E}\left\{[\nabla \tilde{f}_k(x) - \nabla f(x)][\nabla \tilde{f}_k(x) - \nabla f(x)]^\top\right\}$$

$$= \mathbb{E}\left\{[\nabla \tilde{L}_k(x) - \frac{1}{n} \nabla L(x)][\nabla \tilde{L}_k(x) - \frac{1}{n} \nabla L(x)]^\top\right\}$$

$$= \mathbb{E}[\nabla \tilde{L}_k(x) \nabla \tilde{L}_k(x)]^\top - \frac{1}{n^2} \nabla L(x) \nabla L(x)^\top$$

$$= \frac{1}{B}\left[\frac{1}{n} \sum_{i=1}^n \nabla L_i(x) \nabla L_i(x)^\top - \frac{1}{n^2} \nabla L(x) \nabla L(x)^\top\right].$$

## A.2 Transformation of hSGD

By multiplying both sides of hSGD by $Q^\top$ we obtain

$$d(Q^\top X_t) = -\gamma Q^\top \nabla L^{\text{reg}}(X_t) dt + \gamma Q^\top \sqrt{\frac{2}{B} L(X_t) \nabla^2 L(X_t)} \, dW_t$$

$$= -\frac{\gamma}{n} Q^\top \left[A^\top (AX_t - b) + \delta X_t\right] dt + \frac{\gamma}{n} \sqrt{\frac{1}{B} |AX_t - b|^2} Q^\top \sqrt{A^\top A} \, dW_t. \tag{18}$$

Due to

$$Q^\top \left[ A^\top (AX_t - b) + \delta X_t \right] = \left( \Sigma^2 + \delta I_r \right) Q^\top X_t - \Sigma P^\top b$$

and

$$|AX_t - b|^2 = |\Sigma Q^\top X_t - P^\top b|^2, \quad Q^\top \sqrt{A^\top A} = \Sigma Q^\top,$$

(18) can be reformulated as

$$
\begin{aligned}
d(Q^\top X_t) = &- \frac{\gamma}{n} \left[ \left( \Sigma^2 + \delta I_r \right) Q^\top X_t - \Sigma P^\top b \right] dt \\
&+ \frac{\gamma}{n} \sqrt{\frac{1}{B} |\Sigma Q^\top X_t - P^\top b|^2} \Sigma \, d\left( Q^\top W_t \right).
\end{aligned}
\tag{19}
$$

Let $B_t := Q^\top W_t$, which is an $r$-dimensional Brownian motion, due to $Q^\top Q = I_r$. From (19) it follows that $Y_t = Q^\top X_t - Q^\top x_*$ satisfies

$$dY_t = -\frac{\gamma}{n} \left[ \left( \Sigma^2 + \delta I_r \right) Y_t - \alpha \right] dt + \frac{\gamma}{n} \sqrt{\frac{1}{B} [Y_t^\top \Sigma^2 Y_t + \beta]} \Sigma dB_t \tag{20}$$

with

$$\alpha := -\Sigma^{-1} P^\top b, \qquad \beta := b^\top (I_n - P P^\top) b \geq 0.$$

Reading (20) component by component, we obtain (4).

## A.3   Proof of Theorem 3.1

We write the SDEs (7) and (8) in the form

$$dZ_t^i = b_i(Z_t^i)dt + \sigma_i(Z_t)dB_t^i, \quad d\hat{Z}_t^i = b_i(\hat{Z}_t^i)dt + \hat{\sigma}_i(\hat{Z}_t^i)dB_t^i,$$

where

$$b_i(z_i) = -\theta_i(z_i - \mu_i), \quad \sigma_i^2(z) = 2\theta_i \phi_i(|z|^2 + \chi) \quad \text{and} \quad \hat{\sigma}_i(z_i)^2 = 2\theta_i \phi_i(z_i^2 + \chi).$$

While the drift coefficients are identical, the diffusion coefficients satisfy the inequality $\sigma_i(z) \geq \hat{\sigma}_i(z)$ for all $z \in \mathbb{R}^r$ and $i = 1, \ldots, r$. Note that all coefficients are Lipschitz continuous and of bounded growth, such that the standard assumptions for uniqueness and existence of strong SDE solutions are satisfied. Moreover, the SDEs for $\hat{Z}_t^i$ are decoupled and each is a Markov diffusion with generator given by

$$\hat{\mathcal{L}}_i = b_i(x)\partial_x + \frac{\hat{\sigma}_i(x)^2}{2} \partial_{xx},$$

where $x$ denotes the scalar state variable of $\hat{Z}^i$. Let $C_P^l(\mathbb{R})$ denote the subspace of $C^l$-functions for which all derivatives up to order $l$ have polynomial growth. Suppose that $g \in C_P^l(\mathbb{R})$. From Theorem 4.8.6 in [Kloeden and Platen, 1999] the backward functional

$$\mathcal{G}_i(t, x) = \mathbb{E}[g(\hat{Z}_T^i) | \hat{Z}_t^i = x], \quad t \in [0, T],$$

satisfies the backward Kolmogorov equation

$$
\begin{aligned}
\partial_t \mathcal{G}_i(t, x) + \hat{\mathcal{L}}_i \mathcal{G}_i(t, x) &= 0 \quad t < T, \\
\mathcal{G}_i(T, x) &= g(x).
\end{aligned}
\tag{21}
$$

with $\partial_t \mathcal{G}_i$ continuous and $\mathcal{G}_i(t, \cdot) \in C_P^l(\mathbb{R})$ for each $t \in [0, T]$. We now provide a Lemma showing the *propagation-of-order* property of $\hat{Z}$:

**Lemma A.1.** *If $g \in C_P^l(\mathbb{R})$ is convex, so is $\mathcal{G}_i(t, \cdot)$ for all $t \in [0, T]$ and $i = 1, \ldots, r$.*

*Proof.* For better readability we suppress the superscript and subscript $i$ in the SDE

$$d\hat{Z}_t^i = b_i(\hat{Z}_t^i)dt + \hat{\sigma}_i(\hat{Z}_t^i)dB_t^i$$

and consider its Euler-Maruyama approximation

$$\hat{Z}_{K,t_{j+1}} = \hat{Z}_{K,t_i} + b(\hat{Z}_{K,t_j})\Delta t_j + \hat{\sigma}(\hat{Z}_{K,t_j})(B_{t_{j+1}} - B_{t_j})$$

with $t_j = j\frac{T-t}{K} + t$, $j = \{0, 1, \cdots, K\}$ and $\Delta t_j = \frac{T-t}{K} := \Delta$. Using Theorem 9.7.4 in [Kloeden and Platen, 1999] we have

$$\mathcal{G}_K(t,x) = \mathbb{E}[g(\hat{Z}_{K,T})|\hat{Z}_{K,t} = x] \to \mathcal{G}(t,x), \quad t \in [0,T]. \tag{22}$$

Let $\mathcal{A}$ be a transition operator given by

$$\mathcal{A}S = S + \Delta b(S) + \hat{\sigma}(S)W$$

with $W \sim N(0, \Delta)$. We will show that $\mathcal{A}$ satisfies the convex-ordering property

$$\mathbb{E}h(S_1) \leqslant \mathbb{E}h(S_2) \Rightarrow \mathbb{E}h(\mathcal{A}S_1) \leqslant \mathbb{E}h(\mathcal{A}S_2) \tag{23}$$

for any convex function $h(\cdot)$. Let $S_1$, $S_2$ be random vectors which are independent of $W$ and satisfy $\mathbb{E}h(S_1) \leqslant \mathbb{E}h(S_2)$. Due to Strassen's theorem in [Strassen, 1965], we can also assume that $\mathbb{E}(S_2|S_1) = S_1$. It follows from conditional Jensen's inequality that

$$\begin{aligned}
\mathbb{E}h(\mathcal{A}S_2) &= \mathbb{E}h(S_2 + \Delta b(S_2) + \hat{\sigma}(S_2)W) \\
&= \mathbb{E}[\mathbb{E}h(S_2 + \Delta b(S_2) + \hat{\sigma}(S_2)W)|S_1] \\
&\geqslant \mathbb{E}[h(\mathbb{E}(S_2|S_1) + \Delta\mathbb{E}(b(S_2)|S_1) + \mathbb{E}(\hat{\sigma}(S_2)|S_1)W)] \\
&= \mathbb{E}[h(S_1 + \Delta b(S_1) + \mathbb{E}(\hat{\sigma}(S_2)|S_1)W)].
\end{aligned} \tag{24}$$

Here, the linearity of $b(\cdot)$ implies $\mathbb{E}(b(S_2)|S_1) = b(S_1)$. Note that the function $f(x) = \sqrt{x^2 + \chi}$ is convex. Similarly, $\sigma(\cdot)$ is convex. Using conditional Jensen's inequality again gives

$$\varpi(S_1) := \mathbb{E}(\hat{\sigma}(S_2)|S_1) \geqslant \hat{\sigma}(\mathbb{E}(S_2|S_1)) = \hat{\sigma}(S_1). \tag{25}$$

Due to

$$S_1 + \Delta b(S_1) + \mathbb{E}(\hat{\sigma}(S_2)|S_1)W \sim N(\mu, \varpi^2), \quad S_1 + \Delta b(S_1) + \hat{\sigma}(S_1)W \sim N(\mu, \hat{\sigma}^2)$$

with $\mu = \mathbb{E}(S_1 + \Delta b(S_1))$, by Theorem 3.4.7 in [Müller and Stoyan, 2002], (25) implies that

$$\mathbb{E}[h(S_1 + \Delta b(S_1) + \mathbb{E}(\hat{\sigma}(S_2)|S_1)W)] \geqslant \mathbb{E}[h(S_1 + \Delta b(S_1) + \hat{\sigma}(S_1)W)] = \mathbb{E}h(\mathcal{A}S_1).$$

Combined with (24) we have proved the convex-ordering property (23).

By the Markov property of the Euler-Maruyama approximation we have

$$\mathcal{G}_K(t,x) = \mathbb{E}[g(\mathcal{A}^K x)].$$

Let $Z$ be a Bernoulli random variable which takes the value $z_1 \in \mathbb{R}$ with probability $p \in (0,1)$ and the value $z_1 \in \mathbb{R}$ with probability $1 - p$. Then $\mathbb{E}(Z) = pz_1 + (1-p)z_2$. Then we have

$$h(\mathbb{E}(Z)) = h(pz_1 + (1-p)z_2) \leqslant ph(z_1) + (1-p)h(z_2) = \mathbb{E}h(Z).$$

Using the convex-ordering property (23) of the operator $\mathcal{A}$ we obtain

$$\mathcal{G}_K(t, pz_1 + (1-p)z_2) = \mathcal{G}_K(t, \mathbb{E}(Z)) = \mathbb{E}[g(\mathcal{A}^K \mathbb{E}(Z))] \leqslant \mathbb{E}[g(\mathcal{A}^K Z)] = \mathcal{G}_K(t, Z) \tag{26}$$

due to $g$ is convex. Take expectation on both sides of (26) gives

$$\mathcal{G}_K(t, pz_1 + (1-p)z_2) \leqslant \mathbb{E}[\mathcal{G}_K(t, Z)] = p\mathcal{G}_K(t, z_1) + (1-p)\mathcal{G}_K(t, z_2),$$

which means $\mathcal{G}_K(t, \cdot)$ is convex. The approximation property (22) implies the convexity of $\mathcal{G}(t, \cdot)$. $\square$

Next, we need a technical result that shows that each process $\mathcal{G}_i(z, Z_t^i)_{t \in [0,T]}$ is of 'class (D)'.[9]

**Lemma A.2.** *For each* $i = 1, \ldots, r$, *the process* $\mathcal{G}_i(t, Z_t^i)_{t \in [0,T]}$ *is of class* (D).

---

[9]A stochastic process $(X_t)_{t \in I}$ is of class (D), if the set $\{X_\tau : \tau \text{ is } I\text{-valued stopping time}\}$ is uniformly integrable (cf. Definition 4.8 in [Karatzas and Shreve, 2012]).

*Proof.* Since the solution to (8) is a polynomial process (see example 3.6 in [Cuchiero et al., 2012]), it follows from Theorem 3.1 in [Filipović and Larsson, 2016] that

$$\mathcal{G}_i(t, Z_t^i) = \mathbb{E}[g(\hat{Z}_T^i)|\hat{Z}_t^i = Z_t^i] = \exp\{(T-t)G\}\mathrm{P}(Z_t^i),$$

where

$$G = \begin{pmatrix} 0 & \mathrm{g}_0 & 2\times 1\mathrm{g}_1 & 0 & \cdots & & 0 \\ 0 & \mathrm{g}_2 & 2\mathrm{g}_0 & 3\times 2\mathrm{g}_1 & 0 & & \vdots \\ 0 & 0 & 2\,(\mathrm{g}_2+\mathrm{g}_3) & 3\mathrm{g}_0 & \ddots & & 0 \\ 0 & 0 & 0 & 3\,(\mathrm{g}_2+2\mathrm{g}_3) & \ddots & & p(p-1)\mathrm{g}_1 \\ \vdots & & & 0 & \ddots & & p\mathrm{g}_0 \\ 0 & & \cdots & & 0 & p\,(\mathrm{g}_2+(p-1)\mathrm{g}_3) \end{pmatrix}$$

with

$$\mathrm{g}_0 = \theta_i\mu_i, \quad \mathrm{g}_1 = \theta_i\phi_i\chi, \quad \mathrm{g}_2 = -\theta_i, \quad \mathrm{g}_3 = \theta_i\phi_i,$$

and $\mathrm{P}(Z_t^i) = (0, 1, Z_t^i, (Z_t^i)^2, \cdots, (Z_t^i)^p)^\mathrm{T}$. Then there is a constant $C_T$ that depends on $T$ such that

$$|\mathcal{G}_i(t, Z_t^i)| \leqslant C_T(1 + |Z_t^i|^p).$$

Let $\tau_n$ be a localizing sequence for $\mathcal{G}(t, y_t)$. Then we have

$$|\mathcal{G}_i(t \wedge \tau_n, Z_{t\wedge\tau_n}^i)| \leqslant C_T(1 + |Z_{t\wedge\tau_n}^i|^p),$$

which implies

$$|\mathcal{G}_i(t \wedge \tau_n, Z_{t\wedge\tau_n}^i)|^2 \leqslant C_T(1 + |Z_{t\wedge\tau_n}^i|^{2p}). \tag{27}$$

Taking $\mathcal{F}_0$-condition on both sides of (27) gives

$$\begin{aligned} \mathbb{E}\left\{|\mathcal{G}_i(t \wedge \tau_n, Z_{t\wedge\tau_n}^i)|^2\right\} &\leqslant C_T\left(1 + \mathbb{E}|Z_{t\wedge\tau_n}^i|^{2p}\right) \\ &\leqslant C_T\left(1 + \mathbb{E}\left[\sup_n |Z_{t\wedge\tau_n}^i|^{2p}\right]\right) \\ &\leqslant C_T e^{CT}. \end{aligned}$$

Here, the last inequality holds based on Lemma 2.17 in [Cuchiero et al., 2012]. Thus, we complete the proof of this lemma. $\square$

We are now prepared to give the proof of Theorem 3.1.

*Proof.* Let $g$ be a convex function and assume for now that $g \in C_P^2(\mathbb{R})$. Define the local martingale

$$L_t = \int_0^t \partial_x\mathcal{G}_i(s, Z_s^i)\sigma_i(Z_s)dB_s^i$$

. Using Itô's formula in the first step and (21) in the second step, we have

$$\begin{aligned} &\mathcal{G}_i(t, Z_t^i) - \mathcal{G}_i(0, Z_0^i) \\ &= \int_0^t \partial_t\mathcal{G}_i(s, Z_s^i)ds + \int_0^t \left(b_i(Z_t^i)\partial_x + \tfrac{\sigma_i^2(Z_t)}{2}\partial_{xx}\right)\mathcal{G}_i(s, Z_s^i)ds + L_t \\ &= -\int_0^t \hat{\mathcal{L}}_i\mathcal{G}_i(s, Z_s)ds + \int_0^t \left(b_i(Z_t^i)\partial_x + \tfrac{\sigma_i^2(Z_t)}{2}\partial_{xx}\right)\mathcal{G}_i(s, Z_s^i)ds + L_t \\ &= \frac{1}{2}\int_0^t [\sigma_i^2(Z_s) - \hat{\sigma}_i^2(Z_s^i)]\partial_{xx}\mathcal{G}_i(s, Z_s^i)ds + L_t. \end{aligned} \tag{28}$$

By $\mathcal{G}_i(t, \cdot) \in C_P^2(\mathbb{R})$ and Lemma A.1 we obtain $\partial_{xx}\mathcal{G}_i(s, \cdot) \geqslant 0$ for all $i \in \{1, \ldots, d\}$. Thus, due to the ordering of $\sigma_i^2$ and $\hat{\sigma}_i^2$, the first term in the right hand side of (28) is nonnegative. Since $L$ is

a continuous local martingale with zero initial data, it follows that $\mathcal{G}_i(t, Z_t) - \mathcal{G}_i(0, Z_0)$ is a local submartingale.

Let $\tau_n$ be a localizing sequence for $\mathcal{G}_i(t, Z_t)$. For all $t \in [0, T]$, we have

$$\mathcal{G}_i(t \wedge \tau_n, Z_{t \wedge \tau_n}) - \mathcal{G}_i(0, Z_0) \xrightarrow[n \to \infty]{a.s.} \mathcal{G}_i(t, Z_t) - \mathcal{G}_i(0, Z_0). \tag{29}$$

Since $\mathcal{G}_i(t, Z_t)$ is a process of class (D) or locally $L^p$-bounded, $p > 1$, it follows that $\mathcal{G}_i(t \wedge \tau_n, Z_{t \wedge \tau_n}) - \mathcal{G}_i(0, Z_0)$ is uniformly integrable. Combining almost-sure convergence with the uniformly integrable property, it implies that the convergence (29) also takes place in $L^1$, and therefore, $\mathcal{G}_i(t, Z_t) - \mathcal{G}_i(0, Z_0)$ is a submartingale. By taking expectations on both sides of (28) and using the fact that $Z_0 = \hat{Z}_0$, we obtain the comparison result

$$\mathbb{E}g(Z_T^i) = \mathbb{E}\mathcal{G}_i(T, Z_T^i) \geqslant \mathcal{G}(0, Z_0^i) = \mathbb{E}[g(\hat{Z}_T^i)] \tag{30}$$

for all convex $g \in C_P^2(\mathbb{R})$.

Now let $g$ be arbitrary convex function on $\mathbb{R}$. From Theorem 3.1.4 in [Hiriart-Urruty and Lemaréchal, 1996] we can find, for each $n \in \mathbb{N}$ a convex Lipschitz function $\tilde{g}_n$ such that $\tilde{g}_n = g$ in $[-n, n]$ and $\tilde{g}_n \leq g$ in $\mathbb{R} \setminus [-n, n]$. By [Azagra, 2013] we can find further smooth convex functions $g_n \in C_{\text{Lip}}^\infty(\mathbb{R})$ such that $\tilde{g}_n - \frac{1}{n} \leq g_n \leq \tilde{g}_n$ on all of $\mathbb{R}$. It follows that the sequence $g_n$ converges pointwise to $g$ from below. We observe that $C_{\text{Lip}}^\infty(\mathbb{R}) \subset C_P^2(\mathbb{R})$ and equation (11) now follows from (30) by monotone convergence. Finally, equation (12) follows by choosing the convex function $g(z_i) = |z_i|^p$. $\qquad\square$

### A.4 Proof of Theorem 3.2 (upper bound)

From $X_t = QY_t + x_*$, the triangle inequality and the unitary invariance of the Euclidean norm, it follows that $|Y_t| \leq |X_t| + |x_*|$. Thus, we have

$$\frac{\beta^{p/2}}{\lambda_1^p} \mathbb{E}[|Z_t^1|^p] = \mathbb{E}[|Y_t^1|^p] \leq \mathbb{E}[|Y_t|^p] \leq 2^p \left( \mathbb{E}[|X_t|^p] + |x_*|^p \right). \tag{31}$$

Now, let $p > \nu_1$. By Theorem 3.1, Fatou's Lemma, and the properties of the distribution (9) or (10)

$$\limsup_{t \to \infty} \mathbb{E}[|Z_t^1|^p] \geq \liminf_{t \to \infty} \mathbb{E}[|Z_t^1|^p] \geq \liminf_{t \to \infty} \mathbb{E}[|\hat{Z}_t^1|^p] \geq \mathbb{E}[|\hat{Z}_\infty^1|^p] = \infty.$$

Together with (31) this implies that also

$$\limsup_{t \to \infty} \mathbb{E}[|X_t|^p] = \infty$$

and it follows from (5) that the tail index satisfies $\eta \leq p$ for all $p > \nu_1$. Finally, the parameter $\nu_1$ in the limit distribution of $\hat{Z}^1$ is given by $\nu_1 = 1 + \phi_1^{-1}$, where $\phi_1$ can be found in (6). Thus, we obtain Theorem 3.2.

### A.5 Proof of Theorem 3.3 (lower bound)

For better readability, we rewrite (hSGD) in the form

$$dX_t = F(X_t)dt + G(X_t)dB_t \tag{32}$$

with

$$F(X_t) = -\frac{\gamma}{n} \left[ A^{\text{T}}(AX_t - b) + \delta X_t \right], \quad G(X_t) = \frac{\gamma}{n} \sqrt{\frac{1}{B} |AX_t - b|^2 A^{\text{T}} A}.$$

Our goal is to show that

$$\limsup_{|x| \to \infty} \frac{(1 + |x|^2) \left[ 2x^{\text{T}} F(x) + |G(x)|^2 \right] - (2 - \rho)|x^{\text{T}} G(x)|^2}{|x|^4} < -C_1 \tag{33}$$

for all $\rho \in (0, \eta_*)$, where $C_1$ is a positive constant and

$$\eta_* := 1 + \frac{2n(\lambda_1^2 + n\delta)}{\gamma \lambda_1^4} - \frac{\sum_{i=2}^d \lambda_i^2}{\lambda_1^2} > 0.$$

Under condition (33) it follows directly from Theorem 5.2 in [Li et al., 2019] that the solution $X_t$ of the SDE (32)) satisfies

$$\sup_{0 \leqslant t < \infty} \mathbb{E}|X_t|^\rho \leqslant C_2$$

with $C_2$ a positive constant, showing Theorem 3.3.

In order to show (33), let

$$M(x) := \frac{x^{\mathrm{T}} A^{\mathrm{T}} A x}{|x|^2}, \quad x \in \mathbb{R}^d \setminus \{0\}$$

denote the Rayleigh-quotient of $A^{\mathrm{T}} A$. From Chapter 1 in [Horn and Johnson, 2012] we have that the range of $M(x)$ is equal to the line segment $[\lambda_r^2, \lambda_1^2]$, i.e.,

$$\left\{ M(x) : x \in \mathbb{R}^d \setminus \{0\} \right\} = [\lambda_r^2, \lambda_1^2]. \tag{34}$$

Evaluating the condition (33), we have

$$\frac{(1+|x|^2) \left[ 2x^{\mathrm{T}} F(x) \right]}{|x|^4}$$

$$= \frac{(1+|x|^2) \left\{ -2\frac{\gamma}{n} x^{\mathrm{T}} \left[ A^{\mathrm{T}}(Ax - b) + \delta x \right] \right\}}{|x|^4}$$

$$= \frac{(1+|x|^2) \left[ -2\frac{\gamma}{n} x^{\mathrm{T}}(A^{\mathrm{T}} A + \delta I_r)x + 2\frac{\gamma}{n} x^{\mathrm{T}} A^{\mathrm{T}} b \right]}{|x|^4}$$

$$= -\frac{2\frac{\gamma}{n} x^{\mathrm{T}}(A^{\mathrm{T}} A + \delta I_r)x}{|x|^4} - \frac{2\frac{\gamma}{n} x^{\mathrm{T}}(A^{\mathrm{T}} A + \delta I_r)x}{|x|^2} + \frac{2\frac{\gamma}{n}(1+|x|^2)x^{\mathrm{T}} A^{\mathrm{T}} b}{|x|^4}$$

and

$$\frac{(1+|x|^2)|G(x)|^2 - (2-\rho)|x^{\mathrm{T}} G(x)|^2}{|x|^4}$$

$$= \frac{(1+|x|^2) \left[ \frac{\gamma^2}{n^2 B} |\sqrt{|Ax - b|^2 A^{\mathrm{T}} A}|^2 \right] - (2-\rho)\frac{\gamma^2}{n^2 B} |x^{\mathrm{T}} \sqrt{|Ax - b|^2 A^{\mathrm{T}} A}|^2}{|x|^4}$$

$$= \frac{\frac{\gamma^2}{n^2}(1+|x|^2)|Ax - b|^2|\sqrt{A^{\mathrm{T}} A}|^2 - (2-\rho)\frac{\gamma^2}{n^2}|Ax - b|^2|x^{\mathrm{T}} \sqrt{A^{\mathrm{T}} A}|^2}{|x|^4}$$

$$= \frac{\frac{\gamma^2}{n^2 B}|Ax - b|^2|\sqrt{A^{\mathrm{T}} A}|^2}{|x|^4} + \frac{\frac{\gamma^2}{n^2 B}|Ax - b|^2|\sqrt{A^{\mathrm{T}} A}|^2}{|x|^2}$$

$$\quad - \frac{(2-\rho)\frac{\gamma^2}{n^2 B}|Ax - b|^2}{|x|^2} \frac{x^{\mathrm{T}} A^{\mathrm{T}} A x}{|x|^2}.$$

With $|\sqrt{A^{\mathrm{T}} A}|^2 = \operatorname{tr}(A^{\mathrm{T}} A)$ and the positive constant $\rho$ given below, we obtain

$$\limsup_{|x| \to \infty} \frac{(1+|x|^2) \left[ 2x^{\mathrm{T}} F(x) + |G(x)|^2 \right] - (2-\rho)|x^{\mathrm{T}} G(x)|^2}{|x|^4}$$

$$= \limsup_{|x| \to \infty} \left[ -\frac{2\frac{\gamma}{n} x^{\mathrm{T}}(A^{\mathrm{T}} A + \delta I_r)x}{|x|^2} + \frac{\frac{\gamma^2}{n^2 B}|Ax - b|^2|\sqrt{A^{\mathrm{T}} A}|^2}{|x|^2} \right.$$

$$\left. - \frac{(2-\rho)\frac{\gamma^2}{n^2 B}|Ax - b|^2}{|x|^2} \frac{x^{\mathrm{T}} A^{\mathrm{T}} A x}{|x|^2} \right] \tag{35}$$

$$= -\frac{\gamma^2}{n^2 B} \liminf_{|x| \to \infty} \left[ \frac{2nB(M(x) + \delta)}{\gamma} - \operatorname{tr}(A^{\mathrm{T}} A)M(x) + (2-\rho)M(x)^2 \right]$$

$$= -\frac{\gamma^2}{n^2 B} \inf_{m \in [\lambda_r^2, \lambda_1^2]} q(m, \rho),$$

where

$$q(m, \rho) = \frac{2nB(m + \delta)}{\gamma} - \text{tr}(A^{\text{T}}A)m + (2 - \rho)m^2. \tag{36}$$

Set

$$\vartheta := 2 + \frac{2nB(\lambda_1^2 + \delta)}{\gamma\lambda_1^4} - \frac{\sum_{i=1}^{d} \lambda_i^2}{\lambda_1^2}.$$

Note that due to the assumption $\gamma < \bar{\gamma}$ we have $\vartheta > 2$. We claim that

$$\inf_{m \in [\lambda_r^2, \lambda_1^2]} q(m, \rho) > q(\lambda_1^2, \theta) = 0 \tag{37}$$

for all $\rho \in [2, \vartheta)$. First, note that $m \mapsto q(m, \rho)$ is concave for any $\rho \in [2, \vartheta)$, such that its minimum must be attained at one of the boundary values $m \in \{\lambda_r^2, \lambda_1^2\}$. Second, note that $\rho \mapsto q(m, \rho)$ is strictly decreasing for any $m \in (0, \infty)$, such that for (37) it is sufficient to show

$$q(\lambda_r^2, \theta) \geq q(\lambda_1^2, \theta) = 0. \tag{38}$$

Using the assumption $\gamma < \bar{\gamma}$ we obtain

$$q(\lambda_r^2, \theta) = \frac{2nB}{\gamma}(\lambda_r^2 + \delta) - \text{tr}(A^{\text{T}}A)\lambda_r^2 + \frac{2nB}{\gamma}(\lambda_1^2 + \delta)\frac{\lambda_r^4}{\lambda_1^4} - \text{tr}(A^{\text{T}}A)\frac{\lambda_r^4}{\lambda_1^2}$$

$$\geq \text{tr}(A^{\text{T}}A)\left(\frac{(\lambda_r^2 + \delta)}{(\lambda_1^2 + \delta)}\lambda_1^2 - \lambda_r^2\right).$$

For $\delta = 0$ the right hand side vanishes and (38) is shown. Differentiation shows that the right hand side is increasing in $\delta$, such that (38) holds for all $\delta \geq 0$. Altogether, we have shown that the right hand side of (35) is strictly negative. Thus, the SDE (32) satisfies the Assumption 5.1 in [Li et al., 2019]. Based on Theorem 5.2 in Li et al. [2019], the solution $X_t$ of the SDE (32) satisfies

$$\sup_{0 \leqslant t < \infty} \mathbb{E}|X_t|^{\rho} \leqslant C$$

for all $\rho \in [2, \vartheta)$. Therefore, the lower bound, denoted by $\eta_*$, for the asymptotic tail index of $X_t$ is

$$\eta_* = \vartheta = 1 + \frac{2nB(\lambda_1^2 + \delta)}{\gamma\lambda_1^4} - \frac{\sum_{i=2}^{d} \lambda_i^2}{\lambda_1^2}.$$

### A.6 Wasserstein convergence

**Lemma A.3.** *Let $Z$ and $\tilde{Z}$ be two strong solutions of* (7) *with possibly different initial conditions $Z_0, \tilde{Z}_0 \in \mathbb{R}^r$. Suppose that*

$$\gamma < \gamma' =: \frac{nB}{2}\left\{\sum_{i=1}^{r} \frac{\lambda_i^4}{\lambda_i^2 + \delta}\right\}^{-1}. \tag{39}$$

*Then the equation*

$$\sum_{i=1}^{r} \frac{\lambda_i^4}{\lambda_i^2 + \delta - n\rho/\gamma} = \frac{nB}{2\gamma} \tag{40}$$

*has a unique positive solution $\rho_* > 0$ and there exist constants $C, C'$ independent of $Z_0, \tilde{Z}_0$, such that*

$$\mathbb{E}\left[\left|Z_t - \tilde{Z}_t\right|^2\right] \leq Ce^{-2t\rho_*}\left|Z_0 - \tilde{Z}_0\right|^2$$

*and*

$$\mathbb{E}\left[|Z_t|^2\right] \leq C'e^{-2t\rho_*}|Z_0|^2.$$

*Proof.* We set $\mu = (\mu_1, \ldots, \mu_r)$, $\Theta = \text{diag}(\theta_1, \ldots, \theta_r)$, $\psi = (2\phi_1\theta_1, \ldots, 2\phi_r\theta_r)$, and transform $Z$ into $V_t := e^{\Theta t}(Z_t - \mu)$. Applying Ito's formula, we see that $V$ can be written as

$$V_t = Z_0 + \int_0^t e^{\Theta s}\sqrt{\text{diag}(\psi_1, \ldots, \psi_r)(|Z_s|^2 + \chi)}dB_s. \tag{41}$$

The same representation holds for $\tilde{V}$ in relation to $\tilde{Z}$. Setting $d(z, z') = \sqrt{|z|^2 + \chi} - \sqrt{|z'|^2 + \chi}$, we estimate

$$\left| V_t^i - \tilde{V}_t^i \right|^2 \leq 4 \left\{ \left| Z_0^i - \tilde{Z}_0^i \right|^2 + \psi_i \cdot \left( \int_0^t e^{\theta_i s} d(Z_s, \tilde{Z}_s) dB_s^i \right)^2 \right\}$$

for each $i = 1 \ldots r$. Using Ito isometry and the Lipschitz property $d(z, z') \leq |z - z'|$, we obtain

$$\mathbb{E}\left[ \left| V_t^i - \tilde{V}_t^i \right|^2 \right] \leq 4 \left\{ \left| Z_0^i - \tilde{Z}_0^i \right|^2 + \psi_i \int_0^t e^{2\theta_i s} \mathbb{E}\left[ |Z_s - Z_s'|^2 \right] ds \right\}.$$

Introducing $D_t = (D_t^1, \ldots, D_t^r)$, where $D_t^i = \mathbb{E}\left[ \left| Z_t^i - \tilde{Z}_t^i \right|^2 \right]$ and $M = \psi \mathbf{1}\top = (\psi_i)_{i,j}$, where $\mathbf{1} = (1, \ldots, 1)$, we can combine these inequalities into the vector-valued inequality

$$D_t \leq 4 \left\{ e^{-2\Theta t} D_0 + e^{-2\Theta t} \int_0^t e^{2\Theta s} M D_s ds \right\}.$$

Now, consider the comparison equality

$$\hat{D}_t = 4 \left\{ e^{-2\Theta t} D_0 + e^{-2\Theta t} \int_0^t e^{2\Theta s} M \hat{D}_s ds \right\}.$$

Differentiation shows that

$$\frac{d}{dt} \hat{D}_t = -2(\Theta - 2M) \hat{D}_t.$$

Applying the comparison result of Beesack [1969], we obtain

$$\mathbb{E}\left[ |Z_s - Z_s'|^2 \right] = \mathbf{1}^\top D_t \leq \mathbf{1}^\top \hat{D}_t = \mathbf{1}^\top e^{-2t(\Theta - 2M)} D_0.$$

Hence,

$$\mathbb{E}\left[ |Z_s - Z_s'|^2 \right] \leq C e^{-2\rho_* t} |Z_0 - Z_0'|^2$$

, where $\rho_*$ is the smallest Eigenvalue of $\Theta - 2M$.

Now, $M = \psi \mathbf{1}^\top$, i.e., $\Theta - 2M$ can be considered a rank-one perturbation of the diagonal matrix $\Theta$. By [Anderson, 1996], the Eigenvalues $\rho_1, \ldots, \rho_r$ of $\Theta - 2M$ are solutions of the *secular equation*

$$F(\rho) := 1 - \sum_{i=0}^r \frac{2\psi_i}{\theta_i - \rho} = 0. \tag{42}$$

Moreover, they interlace the diagonal values of $\Theta$, i.e., we have $\rho_* = \rho_1 < \theta_1 < \rho_2 < \cdots < \rho_r < \theta_r$. Therefore, all Eigenvalues of $\Theta - 2M$ are positive, except for $\rho_*$ which may be either positive or negative. On $(-\infty, \theta_1)$ the function $F$ is strictly decreasing from 1 to $-\infty$, such that its root $\rho_*$ satisfies $\rho_* > 0$ if and only $F(0) > 0$. Rewriting this condition in terms of (6) yields (39); doing the same for the secular equation (42) yields (40). This completes the proof for the estimate of $\mathbb{E}\left[ |Z_s - Z_s'| \right]^2$; the proof for $\mathbb{E}\left[ |Z_s|^2 \right]$ is completely analogous. □

We are now prepared for the proof of Theorem 3.4, which uses some key ideas from [Friesen et al., 2020]:

*Proof.* Let $(Z_t)_{t \geq 0}$ be the unique strong solution of (7) and denote by $p_t(z, d\zeta)$ its Markov transition kernel. Moreover, for any Borel measure $\mu$ on $\mathbb{R}^r$ set

$$P_t \mu(d\zeta) := \int_{\mathbb{R}^r} p_t(z, d\zeta) \mu(dz).$$

Note that $P_{t+s} = P_t P_s = P_s P_t$ by the Markov property of $Z$. Denote by $\mathcal{P}_2$ the set of all Borel measures $\mu$ on $\mathbb{R}^r$ with $\int |z|^2 \mu(dz) < \infty$. From Lemma A.3 we see that under condition (39) $P_t$ maps $\mathcal{P}_2$ into $\mathcal{P}_2$ for any $t \geq 0$. Moreover, the contraction estimate in Lemma A.3 implies that

$$\mathcal{W}_2(P_t \delta_z, P_t \delta_{z'}) \leq C e^{-t\rho_*} |z - z'|$$

with $\delta_z, \delta_{z'}$ the Dirac measures in $z$ and $z'$ respectively. Using the convexity of the 2-Wasserstein distance (cf. Sec. A.2 in [Friesen et al., 2020]), it now follows that

$$\mathcal{W}_2(P_t\mu, P_t\nu) \le Ce^{-t\rho_*}\mathcal{W}_2(\mu, \nu)$$

for any $\mu, \nu$ in $\mathcal{P}_2$.

Let $\mu \in \mathcal{P}_2$. For any $n, k \in \mathbb{N}_0$, we have

$$\mathcal{W}_2(P_{n+k}\mu, P_n\mu) = \mathcal{W}_2(P_nP_k\mu, P_n\mu) \le Ce^{-n\rho_*}\mathcal{W}_2(P_k\mu, \mu),$$

which shows that $(P_n\mu)_{n\in\mathbb{N}_0}$ is a Cauchy sequence in $(\mathcal{P}_2, \mathcal{W}_2)$. In particular there exists a limit $\pi \in \mathcal{P}_2$ such that $\lim_{n\to\infty}\mathcal{W}_2(P_n\mu, \pi) = 0$. Next, we show that $\pi$ is an invariant measure for $Z$. Indeed, for any $h > 0$ and $k \in \mathbb{N}$, we can estimate

$$\mathcal{W}_2(P_h\pi, \pi) \le \mathcal{W}_2(P_h\pi, P_hP_k\mu) + \mathcal{W}_2(P_kP_h\mu, P_k\mu) + \mathcal{W}_2(P_k\mu, \pi) \le$$
$$\le Ce^{-h\rho_*}\mathcal{W}_2(\pi, P_k\mu) + Ce^{-k\rho_*}\mathcal{W}_2(P_h\mu, \mu) + \mathcal{W}_2(\pi, P_k\mu),$$

where the right hand side tends to zero as $k \to \infty$. Finally, we show that the invariant measure $\pi$ is unique. Suppose that there is another invariant measure $\pi' \in \mathcal{P}_2$. Then

$$\mathcal{W}_2(\pi, \pi') = \mathcal{W}_2(P_n\pi, P_n\pi') \le Ce^{-n\rho_*}\mathcal{W}_2(\pi, \pi'),$$

which tends to zero as $n \to \infty$. Together, this shows that under the conditions of Lemma A.3, $Z$ converges in $\mathcal{W}_2$-distance to its unique invariant distribution $\pi$, and hence completes the proof of Theorem 3.4.

$\square$

## A.7 Parameter Values

Table 4: Parameters used for Figure 1

| Figure 1 | data | $d$ | $K$ | $\gamma$ | $\overline{\gamma}$ | $\delta$ | $B$ | $\lambda_1$ | $\eta_*$ | $\eta^*$ |
|---|---|---|---|---|---|---|---|---|---|---|
| (a), (d), (g) | $\mathcal{X}$ | 200 | 1000 | 0.015 | 0.037 | 0 | 1 | 319.83 | 3.56 | 3.61 |
| (b), (e), (h) | $\mathcal{Y}$ | 64 | 10000 | 0.100 | 0.133 | 0 | 1 | 137.07 | 2.48 | 2.91 |
| (c), (f), (i) | $\mathcal{Z}$ | 200 | 10000 | 0.200 | 0.304 | 0 | 1 | 93.49 | 2.70 | 3.06 |

Table 5: Parameters used for Figure 2

| Figure 2 | data | $d$ | $K$ | $\gamma$ | $\delta$ | $B$ |
|---|---|---|---|---|---|---|
| (a), (d), (g) | $\mathcal{X}$ | 200 | 3000 | 0.1 | 0 | 1 |
| (b), (e), (h) | $\mathcal{Y}$ | 64 | 3000 | 0.1 | 0 | 1 |
| (c), (f), (i) | $\mathcal{Z}$ | 200 | 3000 | 0.1 | 0 | 1 |

## A.8 Experimental configuration

The computing device that we use for calculating our examples includes a single Intel Core i7-10710U CPU with 16GB memory. Our code is available at: `https://github.com/zhezhejiao/hSGD`.

