# OpenReview forum: "Emergence of heavy tails in homogenized stochastic gradient descent"
_NeurIPS.cc/2024/Conference — NeurIPS 2024 poster_

### Official Review · Reviewer_G3NM · 2024-07-01

**Soundness:** 3
**Presentation:** 2
**Contribution:** 3
**Rating:** 6
**Confidence:** 3

**Summary:**

This paper analyses the emergence of heavy-tails in homogenized SGD applied on linear models with quadratic loss. Following other works, the authors model SGD through homogenized SGD and show theoretically and empirically that such an equation converges naturally to heavy-tailed distributions, despite the fact that the noise appearing in the SDE is Gaussian. The proof technique is based on known properties of Student-t-distributions, which appear in the study of similar equations, and a comparison result for moments. This allows the authors to derive analytic upper and lower bounds of the tail index that they were able to successfully confront with numerical experiments. These numerical experiments suggest that Student-t-distributions may better described the distribution of SGD iterates than previously used stable distributions.

**Strengths:**

- Showing that heavy tails can arise even in the presence of Gaussian-type noise is an original contribution, which may illuminate some aspects of the presence of heavy tails in machine learning.

- As opposed to previous works, analytical bounds are derived on the tail-index. These bounds are directly computable from the optimization hyperparameters.

 - The authors confirmed previous experimental findings regarding the emergence of heavy tails in SGD. Moreover, their results suggest that Student-t-distributions could be used to model SGD noise, as opposed to previously used $\alpha$-stable distributions.

**Weaknesses:**

- The paper has no conclusion, it could improve the paper to add a few lines of conclusion to discuss limitations and future works.

 - As it is explained in Section 2.1, the analysis is entirely limited to the case of linear regression with quadratic loss. However, this is never mentioned neither in the title, abstract, or introduction (including the claimed contributions) of the paper. The claim that the paper analyses ``heavy-tailed phenomena in SGD'' may therefore be too strong. It should be mentioned that all results were obtained for linear regression.

 - A weakness related to the previous point is that there should be more discussion on whether this analysis can extend to more complex settings, ie, outside of the linear regression settings, instead of just saying that all models are approximately quadratic near a local minimum. It is understandable that it is hard to extend the theoretical analysis to this setting, but  it would largely enhance the paper to perform similar experiments as in Figure 1, but with a simple non-linear model, like a 2-layers neural network.

**Questions:**

- In the equation just after line 93, what is $D$?

 - Can you elaborate on how big the right-hand side of Equation (12) can be in practice? In particular, the ratio $n/\gamma$ (number of data points over learning rate) seems to be huge. What are the conditions for the tail index to be small (ie, the distribution has very heavy tails)? In particular, it would be beneficial to present the analytical values found in Table 3 in the main part of the paper (they are only presented in the appendix).

 - Theorem 3.4: in order for the 2-Wasserstein distance to make sense, isn't the fact that the tail index is greater than 2 necessary? Does that follow from the assumptions of this theorem?

 - In Figure 1, you compare the Student-t-distribution and $\alpha$-stable distributions to the experimental data. How was the tail-index $\alpha$ (used for the $\alpha$-stable distributions) estimated?

**Other remarks**

 - In the introduction, you mention that some authors argue for a negative correlation between the tail-index and the generalization error. However, a lot of recent works have argued that different behaviors can happen in many settings, see in particular https://arxiv.org/abs/2006.09313, https://arxiv.org/abs/2402.07723 and https://arxiv.org/abs/2301.11885.

**Limitations:**

Several limitations were discussed in the paper. However, it should be clear from the abstract and introduction that the analysis is restricted to linear models. Moreover, a discussion on the potential extensions of the theoretical (and empirical) result to vanilla SGD and non-linear models would be very interesting.

---

> ### Author Rebuttal · Authors · 2024-08-05
>
> We thank you for your review and the comments on our paper. Below we comment on the Weaknesses and answer to the Questions posed in your review:
>
> * **Lack of Conclusions Section**: In our submission we have omitted a Conclusions section due to space limitations. We will include such a section (discussing future work and current limitations) in the revised version, which allows an additional page.
>
> * **Results limited to quadratic loss**: We _do_ mention this limitation in the paper's abstract, writing `... and show in a _regularized regression framework_ that it leads to an asympotically heavy tailed parameter distribution…'. In line58 we again mention the limitation to quadratic loss, and we will also address in the the new limitations section.
>
> * **Results on more complex models**: We have performed experiments on a 2-layer NN, this results will be included in the revised version.
>
> * **Question 1**: We will insert just above the equation in line 93:  `For any given $T>0$ and $D > 0$ there is a $C> 0$, such that...'
>
> * **Question 2**: We think it is difficult to give general conditions leading to a small tail-index in _absolute_ terms, because the leading singular value $\lambda_1$ affecting the tail-index will be highly data-dependent. Instead we provide insight into the _relative change_ of the tail-index when hyper-parameters are modified, see Corollary 3.5. In the current version Table 3 was moved to the appendix due to space constraints; if space allows we will move Table 3 to the main part of the paper in the revised version.
>
> * **Question 3**: Yes, if the conditions of Theorem 3.4. are satisfied, then the tail index must be greater than 2. We will add a remark on this. In fact, it is also possible to show that if $\gamma > \gamma’$ then the tail-index must be smaller than 2. Thanks for pointing this out.
>
> * **Question 4**: In our experiments, we use the Python module statsmodels for qqplots and distribution fitting, which uses a maximum-likelihood-based method to estimate the distributions' parameters.
>
> * **Other Remarks**: Thanks for pointing out these interesting new results. We will make sure to add a remark in the introduction and cite the mentioned papers. We will also mention in the Conclusion that the link between tail-index and generalization error in the hSGD framework warrants further research.
>
> * **Limitations**: We will add a conclusions/limitations section to the revised version. Regarding extensions to non-linear models, see the general author rebuttal above.

---

> > ### Comment · Reviewer_G3NM · 2024-08-09
> > **Thank you for your answer**
> >
> > I thank the reviewer for addressing my concerns. Here are a few remarks.
> >
> >  - Results limited to quadratic loss: Thank you for this clarification. Indeed I had not understood the term ‘regularized regression’ was implicitly referring to the linear case. I still think that using the term ‘regularized linear regression’ could improve the clarity. I do agree that section 2 makes it very clear that the paper is on the linear case, but I would like it to be clearer from the abstract.
> >
> >  - I believe the additional experimental results with 2 layers neural networks will make the paper better.
> >
> > Based on the rebuttal by the authors, I might be willing to increase my score to 6.

---

> > > ### Author Response · Authors · 2024-08-11
> > >
> > > Thank you for your reply and your comments.
> > >
> > > * Yes, we agree, and we will replace 'regularized regression' by 'regularized linear regression' in the abstract.
> > >
> > > * Thanks, we will include them in the revised paper. More details are given in the general Author Rebuttal.
> > >
> > > If we have sufficiently addressed your concerns, any increase in your score would be appreciated. If further clarification is needed, we are available for discussion.

---

> > > > ### Comment · Reviewer_G3NM · 2024-08-12
> > > > **Thanks**
> > > >
> > > > Thank you again for your answer.
> > > >
> > > > I am satisfied with the rebuttal and believe this paper might provide an interesting perspective on heavy tails in machine learning.
> > > > I will increase my score to 6.
> > > >
> > > > Good luck.

---

### Official Review · Reviewer_qGrH · 2024-07-03

**Soundness:** 3
**Presentation:** 4
**Contribution:** 3
**Rating:** 7
**Confidence:** 4

**Summary:**

The paper builds up on the theory of heavy tails in Stochastic Gradient Descent (SGD), clearing some aspects of the distribution of the parameters learned during training. Leveraging a combination of the works on homogeneized SGD and diffusion, the author(s) are able to shed light on the role of noise and the distributional properties of parameters.
More in depth, they relate homogeneized SGD to a "coupled" Pearson diffusion, that can be put in comparison (convex stochastic order) with the standard one. Thanks to this, an explicit description of heavy-tailedness via the asymptotic tail index is obtained. One side of the bound is automatic, the other requires an assumption on the learning rate. After establishing the asymptotic behavior, the speed of convergence to it is explored; subject to an assumption on the learning rate, the author(s) prove that it can happen exponentially fast in Wasserstein distance. To validate their arguments, statistical tests for the parameter distribution are proposed in one case.

**Strengths:**

The overall exposition is clear and to the point.
- On a related note, the structure is pedagogical: my impression is that any reader at any level could take something from spending time on the paper. This is important.
- The observation that gaussian noise is sufficient to have a heavy-tailed distribution is very valuable.
- Asymptotic results are paired with practical convergence at the cost of assumptions that are not absurd.
- The work connects recent works on the distribution of SGD parameters and well-developed theory of diffusions.
- The experiment proposed to validate one of the results is a theoretically grounded suggestion for modelling parameter distribution, and not just a heavy-tailed educated guess.
- The coupled-Pearson diffusion link sheds a new light on the interpretation of the dynamics.
- The idea of using convex stochastic order is simple and effective to obtain just what is needed. Such point is probably the part I appreciated the most.
- In general, a nice and useful read.

**Weaknesses:**

(more of a subtlety) The convex stochastic order bound works for $p\geq 1$ only, Theorem 3.2 does not explicitly state this. One might ask: what if we have a distribution with asymptotic index less than one? Then Theorem 3.1 does not work, but the truth is that for both cases of the Pearson diffusion we have that the tail index is $\nu_i\geq 1$ . Maybe this should be made more explicit? Can you give more clarifications?
- The diffusion approximation is backed on an approximation of the covariance done to match the moments. Crucially, it makes the problem tractable. The works about hSGD, as the author(s) mention themselves in lines 91-97 are subject to "certain assumptions" and "further empirical evidence". This justifies their paper, to the extent that they trust the same setting, as it does not qualify for a full mathematical proof. For the same reason, the present submission misses a set of assumptions, while proving theorems. In general, I agree with this exploratory perspective, but maybe it needs to be mentioned more explicitly and not just briefly stated (related to the note on Limitations just below).
- A proper, explicit discussion of limitations is missing. Section 2.6 is a comparison with literature, section 3.4 is a reconciliation with a specific work. As a reader -- and this is a personal opinion -- I would be very happy to hear why and when your work might/does not work.
- The experiments show only one of the two scenarios (inverse Gamma type distribution is not shown).
- Figure 1, subfigures (g)-(h)-(i) should show upper and lower bounds on the ccdf but have some inconsistency. I understand that this might be due to the experimental setting, but Figure (g) clearly has the lower bound (blue) above the lower upper bound (green), with the true distribution even below. Maybe you can clarify this aspect?
- Theorem 3.4 is not backed by experiments.
- Overall, I would be glad to see more validation of the theory.


#### Typos
These are not weaknesses, but I am adding them here since there is no dedicated box. Please note that some are very pedantic, but I want to be constructive and give you all that I saw.
- The notation for $L, L_{\Omega_k}, L^{\mathrm{reg}}, L^{\mathrm{reg}}_{\Omega_k}$ is possibly confusing: if I look at equations (SGD), (1) and (2) it looks like there is something off, but then checking the original formulation of the works cited it is possible to understand why this is the case. Maybe a clarification sentence would be helpful to the reader.
- Related to the point above, If we also take into account appendix A.1, there we also need to take care of $L, f$. Maybe too many redundant symbols do not help an inexperienced reader.
- (line 74-75) The standard notation for vectors is to take them as columns, the fact that $C(x) = \mathbb{E}[\epsilon(x)^\top\epsilon(x)]$ on the top of my head makes me think it is a scalar. At the same time, I understand that this is a matter of canonical choice.
- (line 100) The matrix $Q$ is stated to be "$r$-by-$d$" but according to $A = P\Sigma Q^\top$ with $\Sigma$ in $r$-by-$r$ it should be the opposite.
- (line 113) is the power in the covariance $\Sigma$ appearing in $\alpha$ right? I have not thouroughly checked this.
- (line 121, Def. 2.2) $\limsup_{t\to\infty}$ is taken, but there is no $t$, also $F(x)$ appears in the equation. I believe you wanted to write the limit supremum of $z\to\infty$, $1 - F(z)$ at the  numerator, and $e^{-sz}$ at the denominator?
- (line 125, Def. 2.4) you take the modulus of a vector in the definition of tail index. Later in equation (6) you take the norm of $Z_t$, where $Z_t$ is a vector. In one way or the other I would like to point out the inconsistency of notation for norms of vectors.
- (Figure 1, subfigs. g, h, i and caption) in the legend, you have $t(\eta_i)$, in the caption, you use the notation $\eta_*, \eta^*$, the two are intuitively bijected, but maybe you can fix the legend to be consistent with the paper?
- secondary, but related to the point above: the legend in the subfigures is very small and redundantly repeated. If you have space left from the submission page limit, and time in these busy times, could you please build a common legend in a separate subfigure? I am guessing here, but it looks like this is Matplotlib, and there are quick ways to extract a legend as a png from a figure, and add it to the text. This is more of an aesthetic aspect, please take it as a random idea rather than a request.
- (lines 419-420) The sterile comment is that the "class (D)" TeX typesetting is wrong, there is a specific command for quotations. The constructive one is that you may want to remove the hat from the $\hat{Z}_t^i$ in line 419
- (lines 446-447) the function $\tilde{g}_n$ is bounded by $g$ in the same way in the two intervals.
- (between line 454-455) in the last inequality, you are missing an $\mathbb{E}$.

**Questions:**

What ensures that the solutions are strong?
- (more of a curiosity) Do you have any comments on a possible regime/example in which the lower bounds on the learning rate $\gamma$ obtained in Theorems 3.3-3.4 give a nice interpretation?
- In the experiments, you analyze the first eigen-component of $\mathbf{Q}$ via the projected quantity $\mathbf{q}_1^\top \mathbf{x}_K$, what happens to the analogues for $\mathbf{q}_2, \ldots$? Do you observe that the theory works less? Maybe it would be interesting to see other experiments.
- Can you somehow validate Theorem 3.4 with an experiment? The parameter $\rho_{\star}$ is the solution of an equation, you have an explicit bound on the learning rate that if verified would make you expect to have a certain exponential convergence, do you see it in a simulation?

The paper is technically solid, and of interest to the community. Subject to fixing the typos and potentially extending the experimental section, I believe it is very good. Therefore, count my score as "waiting for engagement".  Dear author(s), please do. I will be glad to raise my scores.

**Limitations:**

None.

---

> ### Author Rebuttal · Authors · 2024-08-05
>
> We thank you for your extensive comments and for your detailed reading of our paper. Regarding the mentioned Weaknesses, Typos and Questions, our replies are as follows:
>
> * **Case of tail-index < 1**: Our results should be interpreted as follows: By Theorem 3.2. hSGD will _always_ have an asymptotic tail-index of at least one. Thus, if SGD produces (empirically) a distribution with tail-index $\alpha < 1$, it is not Theorem 3.2. which breaks down, but the approximation of SGD by hSGD becomes too weak to track the tail-index. We can add a remark to this effect in the revised version.
>
> * **Limitations**: We have omitted a Limitations/Conclusions section from the submitted version due to space constraints. We will include such a section in the revised version (which allows one additional page). The following Limitations will be addressed: (1) Need for results beyond [1] to control the 'gap' between hSGD and discrete SGD, (2) extension of results to non-quadratic and non-convex losses, (3) Current results cannot be applied to 'extremely heavy tails' with tail index $\alpha < 1$.
>
> * **Small inconsistencies in Figure 1.(g)**: The tail bounds relate to the tail behaviour of distributions, such that a different ordering of cumulative distribution functions in the ‘bulk’ of the distribution is not prohibited. Moreover, numerical effects and finite sample effects can contribute to differences between experiment and theory
>
> * **More validation/theory**: In the paper, our focus was on giving a clear exposition of the theory and the new method (comparison in convex order). Thus, we have limited our experiments to the validation of the tail bounds and the distributional properties. Our view is that Theorem 3.4 is not a ‘main result’ of the paper, but rather provides the justification of applying asymptotic tail bounds to non-asymptotic marginal distributions.
>
> * **Typos**: All typos have been fixed, thanks. Selected comments:
>      - _notation in equations (SGD), (1) and (2)_: As the gradient noise is the difference of the two terms $\nabla L^\mathrm{reg}_{\Omega_k}$ and $\nabla L^\mathrm{reg}$, the regularization terms cancel and $L^\mathrm{reg}$ can be replaced by $L$ in (2); is that what you are referring to? If yes, we will add clarifying remarks.
>     - _notation for covariance matrix $C(x)$_: Agreed, we will treat gradients as column vectors and switch transpose signs accordingly.
>     - _Expression for $\alpha$ in line 113_: We have re-checked and it is correct.
>     - _Notation for vector norms_: Thanks, we will use $\||.\||$ throughout.
>
> * **Strong solutions of SDEs (6) and (7)**: As pointed out in line 388f, both drift and diffusion coefficients are Lipschitz, such that the existence of unique strong solutions follows from standard results, such as Thm. 2.5 in Chapter 5 of [2]
>
> * **Interpretation of learning rate bounds in Theorems 3.3. and 3.4**: One observation is that inserting the upper bound $\overline{\gamma}$ of the learning rate in Theorem 3.3. into (13) yields a lower bound of $\eta_* = 2$. In other words, Theorem 3.3. breaks down exactly for tail-indices $\eta < 2$. Another observation (can be shown similar to the proof of Theorem 3.4) is that if the learning rate satisfies $\gamma > \gamma'$ then the second moment of the hSGD process $X_t$ diverges (i.e. also 2-Wasserstein distance diverges).
>
> * **Extension of experimental section**: On request of other reviewers, we will add an experiment on a non-linear mode (2-layer NN). Adding further experiments, e.g. on Wasserstein convergence or properties of other projections $q_i^\top x_K$, is out of scope both due to time- and space-constraints, we are afraid. But it could be considered in our future work.
>
> We hope that we have addressed your comments in a satisfactory manner and remain available for further discussion.
>
>
> [1] Paquette, C., Paquette, E., Adlam, B., & Pennington, J. (2022). Homogenization of SGD in high-dimensions: Exact dynamics and generalization properties.
>
> [2] Karatzas, Ioannis, and Steven Shreve. Brownian motion and stochastic calculus. Vol. 113. springer, 2014.

---

> > ### Comment · Reviewer_qGrH · 2024-08-08
> > **Rebuttal to rebuttal**
> >
> > Dear author(s),
> > thank you for your explanations; I have carefully read them. I agree with all of them. Let me briefly recap:
> >
> > - tail index: great, I believe the remark is useful if you feel like it is.
> > - limitations: great, much needed, thank you.
> > - figures: understood.
> > - more validation: makes sense.
> > - typos: thank you!
> > - strong solutions: maybe, for students that are bombarded with information, it would be nice to cite such standard results (at least that is my opinion). I know what a strong solution is, I know it appears in SDE theory, but maybe I need to take some time to find the right statemetn for the setting of this paper.
> > - interpretation: got it.
> > - extension: fair enough.
> >
> > Overall, I believe the author(s) have thoroughly addressed my review and will raise my score. Hopefully it gets accepted.
> >
> > Good luck.

---

> > > ### Comment · Reviewer_qGrH · 2024-08-08
> > > **clarification**
> > >
> > > score explanation: I believe it is fair given the neurips indication of scores and confidence. I was undecided between 7/8 and went conservative. Hopefully the other reviewers will contribute. If I happen to read the other official comments by the reviewers (I read up until rebuttals of authors), I might raise to 8. In any case, I am available if you comment further.

---

> > > ### Author Response · Authors · 2024-08-11
> > >
> > > Thank you for your reply, we appreciate your comments and your evaluation of our paper. Regarding the existence of strong SDE solutions, we will add citations to [1] and [2] in the paper.
> > >
> > > [1] Karatzas, Ioannis, and Steven Shreve. Brownian motion and stochastic calculus. Vol. 113. springer, 2014.
> > > [2] Oksendal, Bernt. Stochastic differential equations: an introduction with applications. Springer Science & Business Media, 2013.

---

### Official Review · Reviewer_BR2r · 2024-07-11

**Soundness:** 3
**Presentation:** 3
**Contribution:** 3
**Rating:** 5
**Confidence:** 4

**Summary:**

This paper provides another perspective of the emergence of heavy tails in SGD to the recent literature. Unlike the previous literature, the paper assumes that the SGD can be adequately approximated by homogenized SGD, which is a Brownian-motion driven SDE with a given state-dependent diffusion term.

The paper restricts the study to the quadratic loss.

The tail-index bounds are fully explicit.

The paper introduces a comparison method based on convex stochastic order for homogenized SGD. This allows linking SGD to Pearson diffusions in the literature and then obtain bounds for the tail-index. The results suggest that one can use skew student-$t$-distributions as proxy for parameter distributions in neural networks under SGD, in contrast to the $\alpha$-stable distributions that are commonly used in the literature.

**Strengths:**

(1) The idea of using homogenized SGD as a proxy and then obtain upper and lower bounds for the asymptotic tail-index is new.

(2) The upper and lower bounds are explicit.

(3) It provides a new perspective to a recently popular topic in the literature that is the emergence of heavy tails in SGD.

(4) The analysis is rigorous, and the paper comes up with some comparison method and compares the homogenized SGD with Pearson diffusion.

(5) Wasserstein convergence is also obtained.

**Weaknesses:**

There are a few weaknesses of the paper.

(1) Unlike Gurbuzbalaban et al. (2021) in the literature, the paper does not tackle the tail-index of SGD directly, but only the tail-index of homogenized SGD, which is an approximation of the SGD.

(2) Even though an approximation of the SGD is used, and only the quadratic loss is considered, the results for the tail-index only concern upper and lower bounds. Despite being semi-explicit in Gurbuzbalaban et al. (2021), the tail-index in Gurbuzbalaban et al. (2021) is exact.

(3) Since only the upper and lower bounds are obtained for the tail-index, it is natural to ask whether the analysis can be extended beyond the quadratic loss. This is a very reasonable question because upper and lower bounds should be used when it is impossible to obtain the precise tail-index. One expects that it is impossible to obtain the precise tail-index for the non-quadratic loss. But since comparison method is used in this paper, it is natural to ask whether it is possible to compare the non-quadratic loss with quadratic loss to obtain upper and lower bounds for the tail-index beyond the quadratic loss. In my view, that will make the paper much stronger.

**Questions:**

(1) Your definition in Definition 2.2. for heavy-tailedness  basically says a random variable is heavy-tailed if its tail is heavier than any exponential distribution. But your definition for tail-index in Definition 2.4. is about polynomial decay. I think there is some disconnect here. For example, consider $1-F(x)=e^{-\sqrt{x}}$ for any $x\geq 0$. Then, according to your definition, the tail-index is infinity even though it is heavy-tailed. That makes me wonder whether it is better to change your definition of heavy-tailedness. For example, can you say it is heavy-tailed if the tail-index is finite?

(2) The analysis is for quadratic loss. Since your main results are the upper and lower bounds for the asymptotic tail-index,  would it be possible to extend your analysis beyond the quadratic loss?

(3) It seems you did not provide any discussions on the theoretical result Theorem 3.4. For example, it might be worth mentioning how the convergence rate $\rho_{\ast}$ depends on the model parameters.

(4) You description that Gurbuzbalaban et al. [2021] only describe a phase transition of the asymptotic tail-index $\eta$ from $\eta<2$ to $\eta>2$, without giving quantitative estimates of $\eta$ is not that accurate. There, they obtained a semi-explicit formula for the tail-index $\eta$, and when the input data is Gaussian, the tail-index $\eta$ becomes even more explicit, without the necessity to rely on upper and lower bounds.

(5) In your equations (SME) and (hSGD), the $d$-dimensional Brownian motion is denoted as $W_{t}$. But later, in equation (3), $B_{t}$ is used to denote the $d$-dimensional Brownian motion. If they are the same, it is better to use the same notation. If they are different,
you should make the relation between $W_{t}$ and $B_{t}$ more transparent.

**Limitations:**

I did not see many discussions on the limitations, which should be mentioned in the summary. In fact, there is not even a conclusion section of the paper.

---

> ### Author Rebuttal · Authors · 2024-08-05
>
> Thank you for your comments and your assessment of our paper. To your questions we have the following replies:
>
> **Question (1)**: Our definition of `heavy-tailed' follows the standard definition from the literature on heavy tailed distributions, see e.g. Def. 2.2. in [1]. As we write in l126 a finite tail-index implies heavy-tailedness. The opposite is not true, as evident from your example; several other such examples are known in the literature, e.g. the lognormal distribution.
>
> **Question (2)**: _Thank you, this was an excellent question/suggestion!_ We have looked at the case of a general strongly convex loss with Lipschitz gradient and at least the proof of the lower tail bound can be adapted, i.e. heavy tails can be shown! As similar questions have also been raised by other reviewers we give a longer answer in the authors' rebuttal above.
>
> **Question (3)**:  We agree that it should be possible to determine the dependence of $\rho_*$ on the model parameters by implicit differentiation of the equation in line 222. However, $\rho_*$ is only an upper bound for the rate of convergence, not necessarily the true rate of convergence. Hence, analysing these dependencies may be misleading and we would refrain from doing so in the paper.
>
> **Question (4)**: We are not sure what you mean by `semi-explicit’. Could you clarify and maybe point to the concrete result in Gürbüzbalaban et al. [2021] that you are referring to? Taking your answer into account, we will try to make the comparison of our results to Gürbüzbalaban et al. [2021] more balanced.
>
> **Question (5)**: The two processes $W_t$ and $B_t$ are different Brownian motions related by an orthogonal transformation $Q$ as can be seen in l380 in the appendix. We will add a line in the main part of the paper to clarify.
>
> **Discussion of Limitations**: We have omitted a Conclusions/Limitations section from the submission due to space constraints. We will add such a section in the revised paper (which allows one page more).
>
> We hope that we have addressed your questions in a satisfactory manner and are available for further discussion.
>
> [1] S. Foss, D. Korshunov, S. Zachary. _An Introduction to Heavy-Tailed and Subexponential Distributions_.  Springer 2013

---

> ### Author Response · Authors · 2024-08-12
>
> As the discussion phase is coming to a close, we would appreciate any comments on our rebuttal and, potentially, its impact on your evaluation of our paper. _Thanks!_

---

### Official Review · Reviewer_ya3E · 2024-07-11

**Soundness:** 3
**Presentation:** 3
**Contribution:** 3
**Rating:** 6
**Confidence:** 3

**Summary:**

This manuscript studies homogenized SGD (hSGD) to characterize the tail behavior of SGD iterates for solving the ridge regression problem. By comparing the homogenized diffusion with a known diffusion process called Pearson diffusion, the authors provide a lower bound on the tail index of the iterates of homogenized SGD. They analyze how hyperparameters contribute to this lower bound, offering a qualitative analysis of hyperparameter choices and their impact on the distributional limit of SGD iterates.

Overall, the paper makes a solid contribution and offers a more in-depth analysis of the heavy-tail phenomena in SGD compared to existing work. However, there are a couple of important points that need clarification:

1. Is it the case that the marginals of hSGD and SGD have the same distribution? Theorem 1.3 in [1] suggests this holds for certain quadratic statistics, explained through the concentration of measure. However, if this comparison relies on the concentration of measure, the overall distributions of the two random variables can differ significantly despite overlapping in certain statistics. Therefore, I suggest the authors clarify this aspect.

2. The comparison between the diffusion in (6) and (7) may be somewhat loose, especially given that the ratio between their interaction terms can be high. While the authors use experiments to suggest otherwise, the theoretical analysis does not clearly establish this. Could the authors extend their discussion on how tight or loose this comparison is in high dimensions?

[1] Paquette, C., Paquette, E., Adlam, B., & Pennington, J. (2022). Homogenization of SGD in high-dimensions: Exact dynamics and generalization properties.

**Strengths:**

See the Summary part

**Weaknesses:**

See the Summary part

**Questions:**

See the Summary part

**Limitations:**

I am not convinced that the theoretical results presented in this paper are directly applicable to analyzing the iterates of SGD. Therefore, I am currently leaning towards suggesting a Borderline Reject. I look forward to receiving further clarification, which may lead to a reconsideration of my evaluation.

---

> ### Author Rebuttal · Authors · 2024-08-05
>
> Thank you for your comments and your assessment of our paper. Regarding the two points raised, we have the following reply:
>
> 1. **Difference between hSGD and SGD**: In general, the marginals of hSGD and SGD do not have the same distribution. However, as we mention in line 91f, reference [1] provides quantitative and _non-asymptotic_ approximation guarantees to control the difference between quadratic statistics of the iterates of hSGD and SGD. These quadratic statistics include linear projections as a special case, such that they can be applied e.g. to the projection of $X_t$ onto the dominant singular vector $q_1$ of $A$, which is the relevant direction used to determine the tail-index in Theorems 3.2 and 3.3.
> Moreover, the mentioned drawback (difference of distribution between exact SGD and its approximation) is shared by hSGD with *all* other continuous diffusion approximations, such as the Ornstein-Uhlenbeck/Langevin approximation of [2] and [3] or the alpha-stable OU approach of [4], such that our paper is by far not unique in taking this approximation step.
>
>
> 2. **Comparison between diffusions (6) and (7)**: Yes, we agree that we have no theoretical control over the ‘closeness’ of the diffusions (6) and (7) in general. However, as outlined in section 3.1. the solution of (7) provides a *lower bound in convex order* to the solution of (6), which is precisely what is needed to control the tail-index of (6) from above. Thus, it is primarily the _comparison principle_, not the `closeness' that makes (7) useful in relation to (6).
> Moreover, the experiments in 4.2 (and the performed statistical tests) suggest that the stationary distribution of (7) -- the skew Student-t distribution -- is typically a good proxy for the stationary distribution of (6), notwithstanding the lack of rigorous justification.
>
> We hope that your concerns have been addressed in a satisfactory manner and are available for further clarification.
>
> [1] Paquette, C., Paquette, E., Adlam, B., & Pennington, J. (2022). _Homogenization of SGD in high-dimensions: Exact dynamics and generalization properties._
>
> [2] S. Mandt, M. Hoffman, and D. A. Blei. _A variational analysis of stochastic gradient algorithms_. In International Conference on Learning Representations, 2016.
>
> [3] Qianxiao Li, Cheng Tai, and E Weinan. _Stochastic modified equations and adaptive stochastic gradient algorithms._ In International Conference on Machine Learning, pages 2101–2110. PMLR, 2017.
>
> [4] Umut Simsekli, L. Sagun, and Mert Gurbuzbalaban. _A tail-index analysis of stochastic gradient noise in deep neural networks._ In International Conference on Machine Learning, pages 5287–5837, 2019.

---

> ### Author Response · Authors · 2024-08-12
> **Further clarification needed?**
>
> Dear reviewer  ya3E, in your official review, you have asked for clarification regarding the distributional properties of hSGD and SGD and the comparison of diffusions (6) and (7) in the paper:
>
> > I look forward to receiving further clarification, which may lead to a reconsideration of my evaluation.
>
> Was our rebuttal sufficient for clarification? If needed, we are certainly available for further discussion.

---

> ### Comment · Reviewer_ya3E · 2024-08-12
>
> Thank you very much for the response!
>
> From what I understand, the manuscript establishes the heavy-tailed behavior in homogenized SGD without directly connecting this behavior to SGD itself. As I interpret the proofs of Theorems 3.2 and 3.3, the proof technique cannot be extended to establish that connection when $\eta > 2$ since we only know that homogenized SGD concentrates for quadratic statistics (please correct me if I'm mistaken).
>
> That said, their empirical results align with their theoretical predictions. Therefore, even if there isn't an explicit connection between hSGD marginals and SGD iterations established in the manuscript, leaving this for future work seems reasonable. As long as this point is clearly communicated in the final submission, I would be happy to see this paper at NeurIPS. Consequently, I will raise my score from 4 to 6.

---

### Author Rebuttal · Authors · 2024-08-05

Several reviewers have raised the question if and how our results can be **extended beyond the quadratic loss/to non-linear models**. We think that such an extension is possible along the lines of the method introduced in the paper (and we do have some preliminary results) in the following cases:

(1) For a general (non-convex) smooth loss in the _one-dimensional case_. Here, the stochastic differential equations (6) and (7) will coincide but take a more general form due to the general loss function. Nevertheless, the SDE's stationary distribution and its tail behaviour can be analysed through the SDEs 'speed measure' and 'scale function', see e.g. Chapter V.28 in [1], which in turn can be linked to the loss landscape.

(2) Motivated by the comments of reviewer BR2r in particular, we have looked at the case of a _strongly convex_ loss with _Lipschitz gradient_ (aka $\beta$-smooth) in arbitrary dimension.  Using the standard inequalities (such as the Polyak-Łojasiewicz inequality) satisfied by strongly convex, $\beta$-smooth functions, we can now show that the comparison principle of Theorem 3.1. between hSGD and a Pearson diffusion still holds with $p \ge 2$ for such loss functions and are confident that also Theorem 3.2 can be adapted. This in particular would show that parameter distributions under hSGD are heavy-tailed for all strongly convex, $\beta$-smooth losses.

In the revised version of the paper, we will add a conclusions/future work section where we outline these two possibilities of extending the results.

On the empirical side, we have extended the experiments described in section 4.2. to a simple non-linear model (2-layer neural network) and have plotted the corresponding weight distribution in the second layer as QQ-plots (see attached pdf). As for the linear models, the QQ-plots show that the Student-t-distribution (suggested by our theory) provides a better fit than the alpha-stable distribution that has previously been proposed in the literature.

[1] Rogers, L. Chris G., and David Williams. Diffusions, Markov processes, and martingales: Itô calculus. Vol. 2. Cambridge university press, 2000.

---

### Decision · Program_Chairs · 2024-09-25

**Decision:**

Accept (poster)

**Comment:**

All the reviewers agreed that the paper has significant contributions to heavy tails arising in stochastic optimization. However, they also flagged that the paper is somewhat overselling its contributions since the quadratic loss is not mentioned in early in the paper and there is no formal link to original SGD.

I recommend an acceptance by trusting the authors that they will incorporate all the promised changes one by one, especially

* Mentioning linear regression or quadratic loss starting from the abstract
* Making it crystal clear that the results only hold for hSGD not SGD and mentioning that the formal link is still missing.